# Characterization of the VHH-Fc construct rimteravimab in healthy adults and patients hospitalized for mild-to-moderate COVID-19: Two Phase 1 randomized clinical trials

Ellen Jansen[1,◉], Viki Bockstal[1,◉], Florence Herschke[1,*], Per Olsson Gisleskog[2], Manuela Rinaldi[1], Angélique Boerboom[1], Salah Hadi[3], Natalia Gaibu[4], Michel Moutschen[5], Dominique Tersago[1]

1 ExeVir Bio BV, Ghent, Belgium, 2 POG Pharmacometrics Ltd, London, United Kingdom, 3 Early Development Services, PRA Group BV, a PRA Health Sciences Company, Groningen, The Netherlands, 4 Clinical Republican Hospital "Timofei Mosneaga", ARENSIA Exploratory Medicine, Chisinau, Republic of Moldova, 5 Infectious Diseases and Internal Medicine, Centre hospitalier Universitaire de Liège (CHU), Liège, Belgium

◉ These authors contributed equally to this work.
* fherschke@exevir.com, florence.herschke@gmx.com

## Abstract

### Background

Variable Heavy domain of Heavy chains (VHH) are innovative tools to target unique epitopes, yet few have been developed as heavy chain-only antibodies for clinical use. Rimteravimab (referred to here as XVR011) is a humanized antibody developed for the treatment of mild-to-moderate coronavirus disease 2019 (COVID-19), consisting of two identical VHHs targeting the receptor binding domain (RBD) of the severe acute respiratory syndrome coronavirus 2 (SARS-CoV-2) spike, with a human immunoglobulin (Ig) G1 fragment constant of antibody (Fc), silenced for Fc effector functions. We conducted two Phase 1 studies in healthy volunteers or hospitalized COVID-19 patients to evaluate its safety, tolerability, pharmacokinetics and immunogenicity.

### Methods and findings

A randomized, double-blinded, single-center, placebo-controlled, single ascending dose study was performed in healthy volunteers (Phase 1a, EXEVIR0102, EudraCT 2021-003707-17), in parallel to an open-label, multi-center, single ascending dose study in patients hospitalized for mild to moderate COVID-19 (Phase 1b, EXE-VIR0101, EudraCT 2020-005299-36, NCT04884295). Participants received a single intravenous infusion of 250, 500 or 1,000 mg of XVR011. The primary objective for both trials was the safety and tolerability of XVR011. Pharmacokinetics were evaluated as a secondary objective in Phase 1a and as an exploratory objective in Phase

**Data availability statement:** The raw safety, efficacy, and immunogenicity data behind the figures and tables are presented in the supplementary information. The pharmacokinetics data will be made available by sending a request to info@exevir.com, to ensure the continuous protection of personal data.

**Funding:** This work was financially supported by VLAIO (Agentschap Innoveren & Ondernemen, https://www.vlaio.be/en), Project HBC.2020.3083; by Service Public de Wallonie-Recherche (https://recherche.wallonie.be/en/home.html), Project 8495; by the European Commission Horizon 2020 Framework Programme (https://research-and-innovation.ec.europa.eu/funding/funding-opportunities/funding-programmes-and-open-calls/horizon-2020_en), Project 101045949 — XVR011 Phase 2; and by the sponsor ExeVir Bio BV. Apart from the sponsor ExeVir Bio BV, the funders had no role in study design, data collection and analysis, decision to publish, or preparation of the manuscript.

**Competing interests:** I have read the journal's policy and the authors of this manuscript have the following competing interests: EJ, VB, FH, AB, MR, and DT are/were employees of ExeVir Bio at the time of the work presented in this manuscript. POG, SH, NG, and MM were contracted by Exevir to perform the work presented in this manuscript.

**Abbreviations:** ACE2, angiotensin-converting enzyme 2; ADA, anti-drug antibodies; ADCC, antibody-dependent cell-mediated cytotoxicity; ADE, antibody-dependent enhancement; AST, aspartate aminotransferase; CAR-T, chimeric antigen receptor-T; CDC, complement-dependent cytotoxicity; CDRs, complementarity-determining regions; COVID-19, coronavirus disease 2019; FAS, Full Analysis Set; IDMC, independent data monitoring committee; ICH-GCP, International Council for Harmonization for Good Clinical Practice; Ig, immunoglobulin; ITT, intention-to-treat; PD-L1, anti-programmed cell death ligand 1; PK, pharmacokinetics; RBD, receptor binding domain; RT-qPCR, reverse transcription and quantitative polymerase chain reaction; SARS-CoV-2,

1b. Efficacy (evaluated as respiratory parameters and COVID-19 clinical status) and antiviral activity in patients were evaluated as a secondary objective in Phase 1b. Immunogenicity was evaluated as an exploratory objective. Part 2 of the EXE-VIR0101 study (initially a phase 1b/2 study) was not conducted due to the loss of XVR011 potency against SARS-CoV-2 Omicron BA.2. Demographics, safety, efficacy, and immunogenicity were analyzed using descriptive statistics, while pharmacokinetics were analyzed with noncompartmental pharmacokinetics (PK) modeling.

In the Phase 1a study, there were no infusion-related reactions, serious treatment-emergent adverse events (TEAEs) or TEAEs grade ≥3. 22/30 volunteers (73.3%) reported 53 TEAEs (49 Grade 1, 4 Grade 2) with none being related to XVR011. The most common TEAE was headache ($n = 8$, 26.7%) in various treatment groups. In the Phase 1b study, 27 hospitalized patients were enrolled, and followed up to 30 days. Seven patients (25.9%) reported a total of 15 TEAEs, the majority (80%) being mild to moderate (Grade 1–2). There were no treatment-related serious TEAEs. All TEAEs resolved by the end of the study. Peak exposure (maximal concentration, $C_{max}$) and systemic exposure (area under the curve, $AUC_{0-t}$, and $AUC_{0-inf}$) for XVR011 increased dose-proportionally. Geomean half-life ranged from 15.4 to 17.0 days in Phase 1a, while individual half-life ranged from 11.4 to 15.6 days in Phase 1b. SARS-CoV-2 viral load, as detected in nasopharyngeal samples by reverse transcription and quantitative polymerase chain reaction (RT-qPCR), decreased similarly in all cohorts compared to baseline. No treatment-induced anti-drug antibodies (ADA) were detected in Phase 1a. In Phase 1b, higher XVR011 concentrations increased the likelihood of ADA formation, without impacting pharmacokinetics and pharmacodynamics. No obvious dose-response in COVID-19 clinical status or respiratory parameters was observed. Technological limitations included study size, absence of placebo for the Phase 1b, absence of repeated dosing, evolving SARS-CoV-2 variants and standard-of-care.

## Conclusions

XVR011 displayed a favourable safety, tolerability, pharmacokinetics, and immunogenicity profile, both in healthy volunteers and in patients hospitalized for mild to moderate COVID-19. These data pave the way for the design and clinical development of VHH-Fc constructs.

## Author summary
### Why was this study done?

- Conventional antibodies with a heavy chain and light chain have so far been the gold standard for designing therapeutic antibodies.
- Heavy chain-only antibodies hold several powerful advantages over conventional antibodies, including a smaller size and a longer loop to make contact

severe acute respiratory syndrome coronavirus 2; SOC, standard of care; SPEAD, solid phase extraction and acid dissociation; TEAEs, treatment-emergent adverse events; TNFα, anti-tumor necrosis factor alpha; TTSD, Time to COVID-19 related symptoms disappearance; TTHD, The median time to hospital discharge; ULN, upper limit of normal; VHH, Variable Heavy domain of Heavy chains; WHO, World Health Organization.

with the target epitope, which facilitates access to highly conserved epitopes that are very difficult, if not impossible, to reach for the much larger conventional antibodies.

- Here, a heavy chain-only antibody was tested in the clinic in healthy volunteers and patients hospitalized with mild to moderate COVID-19 symptoms, targeting a conserved epitope on the SARS-CoV-2 virus.

## What did the researchers do and find?

- The safety, tolerability, pharmacokinetics, and immunogenicity of the heavy chain-only antibody XVR011 were evaluated in 2 Phase 1 studies, a Phase 1a in 30 healthy volunteers and a Phase 1b in 27 hospitalized COVID-19 patients, via intravenous (IV) administration in 3 evenly assigned dose cohorts (250, 500 and 1,000 mg).

- XVR011 was safe and well tolerated, with only mild to moderate treatment-related adverse events and no infusion site reactions or hypersensitivity reactions.

- The observed antibody half-life was within the expected range for a heavy chain-only antibody. Only 9 out of 27 COVID-19 patients, and no healthy individuals, displayed treatment-induced anti-drug antibodies (ADAs).

## What do these findings mean?

- These studies provide clinical data for an antiviral VHH-Fc antibody and indicate a favorable safety, tolerability, pharmacokinetic, and immunogenicity profile in both healthy and infected individuals, supportive of the further clinical development of humanized VHH-Fc antibodies against respiratory targets (and in particular COVID-19).

- Given that XVR011 lost its potency against the SARS-CoV-2 Omicron BA.2 variant, clinical development of this molecule will not be further pursued.

- Technological limitations of the studies, including study size and absence of repeated dosing, may affect the interpretation of the safety, tolerability, pharmacokinetic, and immunogenicity signals.

## Introduction

Since the World Health Organization (WHO) declared a public health emergency in January 2020, more than 780 million confirmed cases of COVID-19 have been reported worldwide and 7.1 million people have died [1,2]. Advanced age, underlying medical conditions, certain disabilities, being immunocompromised and being unvaccinated or not being up to date on COVID-19 vaccinations are among the known risk factors for severe COVID-19 outcomes [3,4]. Vaccines for COVID-19 have been approved in many countries and have played a substantial role in mitigating the impact of the pandemic by protecting against severe disease, hospitalization,

and death [5]. Complementary to vaccines, therapeutic antibodies such as cilgavimab + tixagevimab, sotrovimab, casirivimab + imdevimab, bamlanivimab, bebtelovimab, or pemivibart, have proven useful in protecting and/or treating individuals at the highest risk of developing severe disease and adverse outcomes, such as older adults and immunocompromised individuals [6–9]. Most vaccines and antibody-based therapeutics target the large envelope spike protein of the severe acute respiratory syndrome coronavirus 2 (SARS-CoV-2), which mediates viral entry into the host cell by engaging with the angiotensin converting enzyme 2 target receptor and catalyzing membrane fusion [6,10]. Since the start of the pandemic, however, SARS-CoV-2 has continued to evolve to escape human immunity. New genetic variants have emerged with acquired mutations in the RBD of the spike protein, enabling them to resist naturally to acquired and drug- or vaccine-induced antibody neutralization [11,12]. Efforts to develop broadly neutralizing antibodies that bind to conserved regions of the SARS-CoV-2 spike protein are therefore warranted to help protect against disease caused by the currently circulating and newly emerging variants, and potentially against future coronavirus outbreaks [13–15].

Compared to conventional antibodies, camelid-derived VHH antibodies are smaller (approximately 15 kDa versus 150 kDa) and typically have longer complementarity determining region (CDR) 3 loops, enabling them to target unique and occluded epitopes that are often highly conserved. They have favorable solubility, stability, and biodistribution profiles that support rapid and potentially better tissue penetration [16,17]. Fusing VHH to a human IgG1 Fc allows customization to meet specific needs, like half-life extension for a longer duration of protection and/or Fc silencing for specific safety reasons. To our knowledge, four VHH-based therapeutics have been launched broadly to date: (i) bivalent von Willebrand factor-targeting VHH caplacizumab to treat thrombotic thrombocytopenic purpura [18], (ii) trivalent VHH ozoralizumab, featuring two anti-tumor necrosis factor alpha (TNFα) VHH and one anti-albumin VHH for rheumatoid arthritis [19,20], (iii) chimeric antigen receptor-T (CAR-T) therapy ciltacabtagene autoleucel, where the CAR protein features two BCMA-targeting VHH, a 4-1BB co-stimulatory domain, and a CD3-zeta signaling cytoplasmic domain, to treat relapsed/refractory multiple myeloma [21], and (iv) envafolimab, an anti-programmed cell death ligand 1 (PD-L1) VHH fused to a human IgG1 Fc, to treat solid tumors [22–24]. While the latter was tested in patients, no VHH-Fc construct has been tested to date in healthy volunteers or in the infectious disease field in general. Here, we studied XVR011, a monospecific VHH-Fc fusion antibody, in healthy volunteers and in patients with coronavirus disease 2019 (COVID-19).

VHH antibodies are being developed as SARS-CoV-2- antibody-based therapeutics, with the goal to access different, potentially more conserved, epitopes on the spike compared to conventional antibodies. VHH72, the VHH used to construct XVR011, was isolated from a llama immunized with prefusion-stabilized MERS and SARS-CoV-1 spike proteins. It binds to an epitope in the RBD of the spike protein that is difficult to access for human antibodies. At the start of the pandemic, VHH72 was found to also neutralize SARS-CoV-2 with high potency and re-purposed for the development of XVR011 [25]. XVR011 is a bivalent humanized antibody construct, consisting of 2 identical VHH72 building blocks fused to a human Fc-fragment of a human immunoglobulin G1. A L234A – L235A substitution (LALA) mutation is engineered in the human Fc [26,27] to strongly reduce the binding to Fc-gamma receptors on immune cells and complement protein C1q, responsible for triggering antibody effector functions like antibody-dependent cell-mediated cytotoxicity (ADCC) and complement-dependent cytotoxicity (CDC). By silencing these interactions, an optimal safety profile was envisioned when this antibody was developed at the start of the COVID-19 pandemic, as there was a general theoretical concern for antibody-dependent enhancement (ADE) of COVID-19 disease at the time [25,28]. The neutralizing activity of XVR011 is based on a dual mechanism (Table A in S1 Appendix), i.e., XVR011 binding to the spike RBD prevents the virus from interacting with the host angiotensin-converting enzyme 2 (ACE2) receptor through steric hindrance and simultaneously trapping the RBD in the destabilizing "up" conformation, causing shedding of the S1 subunit. By fusing VHH72 building blocks to an Fc fragment with a partially silenced effector function, XVR011 might avoid any potential antibody-dependent exacerbation of COVID-19, relying solely on a neutralization mode of action [29]. Here, we report on two clinical studies, a Phase 1a in healthy adults (EXEVIR0102, EudraCT 2021-003707-17) and Phase 1b in adult patients hospitalized with

mild to moderate COVID-19 (EXEVIR0101, NCT04884295), aiming to explore the safety, tolerability, pharmacokinetics, efficacy, and immunogenicity profiles of XVR011.

## Methods

This study is reported as per the CONSORT 2025 and CONSORT-CONSERVE guidelines (S1, S2 Checklists).

### Study ethics

The following local and/or national independent ethics committees/ institutional review boards approved the study proto-cols and informed consent forms (and their amendments) for both studies: Commissie Voor Medische Ethiek UZ Gent, Comité d'éthique Hospitalo-Facultaire Universitaire de Liège, AZ Sint-Maarten – Commissie Ethiek vzw Emmaüs, Comitetul Național de Expertiză Etică a Studiului Clinic, Comitato Etico Milano Area, Comitato Etico Dell'Instituto Nazionale Per Le Malattie Infettive, IEC of the Foundation Beoordeling Ethiek Biomedisch Onderzoek (Approval numbers in Table B in S1 Appendix). Both studies were conducted as outlined in the protocol, in accordance with the Declaration of Helsinki, the International Council for Harmonization for Good Clinical Practice (ICH-GCP) Guideline E6 (R2), and local ethical and legal requirements.

All participants provided written informed consent before study entry.

### Study design and procedures

EXEVIR0102 (EudraCT 2021-003707-17; Fig A in S1 Appendix; S1 Protocol) was a Phase 1a, randomized, double-blind, placebo-controlled single ascending dose study to evaluate the safety and pharmacokinetics of XVR011. Healthy volunteers, recruited from one study site in the Netherlands (PRA Health Sciences, Groningen), between 5 August and 1 October 2021. received one of three XVR011 dose levels (250, 500 and 1,000 mg) or placebo (0.9% (w/v) sodium chloride sterile solution), administered intravenously (IV). Male and nonpregnant, nonlactating female volunteers aged 18–65 years in good physical and mental health were eligible for the study. Volunteers had to be fully vaccinated against COVID-19 at least 14 days prior to study drug administration, or previously had COVID-19 (not within 2 months prior to the date of screening). Other eligibility criteria focused on past or current use of medications or medical disorders that could potentially confound study results or interfere with safe completion of the study. Use of any medication was prohibited during the study, except for paracetamol and hormonal contraceptives. Details on screening and randomization procedures can be found in S1 Appendix.

EXEVIR0101 (ClinicalTrials.gov NCT04884295, EudraCT 2020-005299-36 https://www.clinicaltrialsregister.eu/ctr-search/trial/2020-005299-36/results; Fig A in S1 Appendix; S2 Protocol) was a 2-part clinical study with XVR011 in COVID-19 patients, including an open-label, single ascending dose Phase 1b study (Part 1) to evaluate the safety of 3 XVR011 doses and identify the recommended Phase 2 dose, followed by a randomized double-blind part (Part 2) to evaluate the efficacy and safety of XVR011 compared to placebo (0.9% (w/v) sodium chloride sterile solution). Results of Part 1 of this study in hospitalized patients are presented in this manuscript. 2 study sites in Belgium and the Republic of Moldova enrolled patients between 26 August 2021 and 17 February 2022. The study population included male and nonpregnant, nonlactating female patients aged 18 years and older with an ongoing SARS-CoV-2 infection as confirmed by reverse transcriptase quantitative polymerase chain reaction and/or positive antigen test, and with onset of COVID-19 symptom(s) within 8 days prior to screening or within 2 days prior to screening for dyspnea and/or tachypnea. Only patients requiring hospitalization for COVID-19 medical care and who had a resting oxygen saturation >91% (compensated) were eligible. Patients requiring mechanical ventilation or intensive care treatment were not eligible. Patients were excluded if they received any concomitant treatment with medicinal products with potential or demonstrated anti-SARS-CoV-2 activity, including but not limited to remdesivir, within 30 days prior to study drug administration or any vaccine

against SARS-CoV-2 within 14 days prior to study drug administration. Other eligibility criteria focused on past or current use of medications or medical disorders that could potentially confound study results or interfere with safe completion of the study. Details on randomization procedure, independent data monitoring committee (IDMC) and study protocol amendments can be found in Supplementary Information (S2 Protocol).

On Day 1, within 36 hours of admission, XVR011 was administered as a single IV infusion under the supervision of the investigator or designated person, in addition to COVID-19 standard of care supportive treatment. Patients were closely monitored during and until 2 hours after the infusion and remained hospitalized until at least 72 hours after the infusion (Day 4). Thereafter, patients whose clinical condition had improved sufficiently could be discharged from the hospital. Study assessments were performed daily up to Day 4 in the hospital and thereafter on Days 8, 15, and 29 (end of study) either as inpatients or as outpatients (Fig A in S1 Appendix).

In March 2022, the study was prematurely terminated before Part 2 was initiated due to the loss of neutralization potency by XVR011 against the Omicron variant BA.2 that was spreading worldwide at the time (Table A in S1 Appendix). This decision was not based on emerging safety issues.

### Safety assessment

The primary outcome of the Phase 1a and Phase 1b studies was to evaluate the safety and tolerability (*i.e.*, examination of infusion site reactions) of a single IV infusion of XVR011 at 3 dose levels in healthy volunteers and COVID-19 patients, respectively. Safety evaluation included all treatment-emergent adverse events (TEAEs), severe TEAEs (*i.e.*, TEAEs with severity grade ≥3), serious TEAEs, and adverse events of special interest (*i.e.*, infusion-related reactions and hypersensitivity reactions occurring within 24 hours of study drug administration), clinical laboratory assessments, vital signs, electrocardiograms, and physical examinations. Safety was monitored from the date of signing the informed consent form until the end of the study (Fig A in S1 Appendix, S1, S2 protocols). AEs (adverse events) were coded using Medical Dictionary for Regulatory Activities (version 23.0 or higher) terms. Assignment of treatment-emergent AE causality took into account: (i) the temporal relationship to XVR011 administration; (ii) the possible mechanism-of-action association with the study drug; and (iii) whether there may be a plausible explanation due to underlying disease or comorbidities or concomitant medications with the AE as a known side effect.

### Pharmacokinetic assessment

The PK of a single IV XVR011 dose in healthy volunteers and COVID-19 patients was evaluated as a secondary outcome in Phase 1a Study and as an exploratory objective in Phase 1b study. Blood samples were taken to determine XVR011 concentrations in serum samples of participants who received XVR011 at the timepoints indicated in Fig A in S1 Appendix (S1, S2 protocols). Serum concentrations were determined using a validated electrochemiluminescence assay (Resolian, UK) based on the guidelines on Bioanalytical Method Validation (Reference FDA guideline). Details on the experimental procedure can be found in S1 Appendix.

### Efficacy assessment

The efficacy of a single IV XVR011 dose in patients hospitalized with mild to moderate COVID-19 was evaluated as a secondary outcome in the Phase 1b study. SARS-CoV-2 viral load from nasopharyngeal swabs was assessed with RT-qPCR. Clinical status was assessed using an 8-point ordinal scale as in Study NCT04280705 (Beigel and colleagues, 2020 [30]). COVID-19 patient-related outcomes were evaluated with a 14-questions questionnaire. Respiratory parameters were described as follows: oxygen supplementation parameters (type, start & stop dates, and flow rate), respiratory rate, and oxygen saturation (on room air and with oxygen supplementation). It should be noted that the study was not powered to detect statistically significant differences from baseline and across cohorts, and that no placebo control was included.

## Immunogenicity assessment

The immunogenicity of a single IV XVR011 dose in healthy volunteers and COVID-19 patients was evaluated as an exploratory outcome in Phase 1a and 1b studies. Blood samples were harvested to determine the presence of ADAs in serum samples of participants who received XVR011 at the timepoints indicated in Fig A in S1 Appendix and S1, S2 protocols.

Determination of ADA consisted of 3 sequential steps, i.e., screening, confirmation, and titer assays.

The presence of anti-XVR011 antibodies in human serum samples was determined using a validated solid phase extraction and acid dissociation (SPEAD) method (Resolian, UK) based on the guideline on Immunogenicity Assessment of Biotechnology-Derived Therapeutic Proteins (Reference Europe, Middle East and Africa, EMEA). Details on the experimental procedure can be found in S1 Appendix.

## Statistical analysis

No formal sample size calculations were performed. Considering XVR011 targets a viral protein and does not bind to any human proteins in preclinical studies, 9–10 participants per dose cohort were considered sufficient to perform an initial assessment of the safety of a single IV dose of XVR011.

Descriptive statistics were used to summarize demographic, safety, and immunogenicity data by treatment. Placebo data in the Phase 1a Study were pooled. Demographics, were presented for the Safety Set (i.e., modified ITT where dosing errors are assigned to the dose actually received) of Phase 1a Study and for the Full Analysis Set (FAS) (i.e., all participants to whom study drug had been dispensed, following intention-to-treat (ITT) principles) of Phase 1b study. For both studies, immunogenicity and efficacy (for the Phase 1b) were presented for the FAS and safety data is presented for the Safety Set. Pharmacokinetic analyses were performed on the PK Set (i.e., Safety Set for which at least one time point has measurable XVR011), using a noncompartmental analysis from the XVR011 serum concentration-time data. Details on the PK Sets for both studies can be found in S1 Appendix.

## Study limitations

The technological limitations of the studies presented here that may impact interpretation of safety, PK, pharmacodynamics (PD), and immunogenicity signals include: study size, antigenic shift from SARS-CoV-2 Delta to Omicron that occurred during the course of the Phase 1b study, leading to an early study termination and short follow-up, as well as the absence of repeated dosing. The absence of placebo in the phase 1b, changing standard of care for COVID-19 treatment over time and variable duration since last vaccination are factors potentially impacting the interpretation of the PD results.

## Results

Two Phase 1 studies were conducted for XVR011, the first, to our knowledge, VHH-Fc was tested in a Phase 1a in healthy adults and a Phase 1b in COVID-19 patients hospitalized with mild to moderate symptoms. In both studies, 3 dose levels were evaluated: 250, 500 and 1,000 mg. These dose levels were selected based on PK/PD modeling, derived from in vivo hamster efficacy studies, in vitro potency, and the preclinical PK profile in nonhuman primates (Table D in S1 Appendix, Fig B in S1 Appendix).

### Participant disposition and demographics

The demographic characteristics for both studies are summarized in Table 1.

In the Phase 1a study, 30 healthy volunteers were enrolled and evenly allocated to the 3 successive dose cohorts (250, 500 and 1,000 mg), randomized to receive either XVR011 ($n = 24$) or placebo ($n = 6$). They were enrolled between 5 August and 1 October 2021. All 30 volunteers completed the study per protocol (Fig 1). In total, 16 women (53.3%) and 14 men (46.7%) participated with a mean age of 34.4 ± 14.48 years (range 20–63 years).

PLOS Medicine

**Table 1. Summary of demographic characteristics.**

| Parameter | | EXEVIR0102 – Healthy volunteers | | | | | EXEVIR0101 – Hospitalized COVID-19 patients | | | |
|---|---|---|---|---|---|---|---|---|---|---|
| | | Placebo (*N*=6) | Cohort 1 250 mg (*N*=8) | Cohort 2 500 mg (*N*=8) | Cohort 3 1,000 mg (*N*=8) | Overall (*N*=30) | Cohort 1 250 mg (*N*=9) | Cohort 2 500 mg (*N*=9) | Cohort 3 1,000 mg (*N*=9) | Overall (*N*=27) |
| Age (year) | Mean (SD) | 28.8 (11.9) | 35.0 (16.0) | 37.4 (17.4) | 35.1 (13.0) | 34.4 (14.5) | 60.9 (7.6) | 53.7 (20.1) | 57.2 (13.3) | 57.3 (14.3) |
| | Median | 22.5 | 25.0 | 29.0 | 31.0 | 25.0 | 57.0 | 64.0 | 61.0 | 61.0 |
| | Min; Max | 20; 50 | 21; 60 | 22; 63 | 23; 59 | 20; 63 | 53; 73 | 26; 72 | 27; 70 | 26; 73 |
| Gender (*n* [%]) | Female | 4 (66.7) | 6 (75.0) | 3 (37.5) | 3 (37.5) | 16 (53.3) | 6 (66.7) | 4 (44.4) | 6 (66.7) | 16 (59.3) |
| | Male | 2 (33.3) | 2 (25.0) | 5 (62.5) | 5 (62.5) | 14 (46.7) | 3 (33.3) | 5 (55.6) | 3 (33.3) | 11 (40.7) |
| Race (*n* [%]) | Asian | 0 (0) | 0 (0) | 0 (0) | 1 (12.5) | 1 (3.3) | 0 (0) | 0 (0) | 0 (0) | 0 (0) |
| | Multiple | 0 (0) | 1 (12.5) | 0 (0) | 0 (0) | 1 (3.3) | 0 (0) | 0 (0) | 0 (0) | 0 (0) |
| | Other | 1 (16.7) | 0 (0) | 0 (0) | 0 (0) | 1 (3.3) | 0 (0) | 0 (0) | 0 (0) | 0 (0) |
| | White | 5 (83.3) | 7 (87.5) | 8 (100) | 7 (87.5) | 27 (90.0) | 9 (100) | 9 (100) | 9 (100) | 27 (100) |
| Ethnicity (*n* [%]) | Not Hispanic or Latino | 6 (100) | 8 (100) | 8 (100) | 8 (100) | 30 (100) | 9 (100) | 9 (100) | 9 (100) | 27 (100) |

Max = maximum; Min = minimum; N = number of participants included in the analysis set; n = number of participants with this characteristic; SD = standard deviation.

Descriptive statistics for demographic data are presented for participants included in the FAS.

In the Phase 1b study, 27 patients were enrolled and evenly allocated to 3 successive dose cohorts (250, 500 and 1,000 mg) (Fig 1). One patient assigned to the 250 mg cohort erroneously received the 500 mg dose and was subsequently included as part of the 250 mg cohort in the full analysis set (FAS) and as part of the 500 mg cohort in the Safety Set. All patients in the 250 and 500 mg cohorts completed the study per protocol. One participant in the 1,000 mg cohort withdrew consent and discontinued the study 18 days after XVR011 administration. In total, 16 women (59.3%) and 11 men (40.7%) participated with a mean age of 57.3 ± 14.33 years (range 26–73 years). Patients were recruited from 2 study sites in Belgium (CHU Liège, 3 patients) and Moldova (Clinical Republican Hospital "Timofei Mosneaga", 24 patients) between 26 August 2021 and 17 February 2022.

Of all patients who enrolled in the Phase 1b study at Day −1, 26/27 (96.3%) were still SARS-CoV-2 positive at Day 1 and 8 when the variant was determined. In the 250 mg cohort group, 8 patients (88.9%) were infected with the Delta 21J variant and 1 (11.1%) with the Delta 21A variant. In the 500 mg cohort group, 6 patients were infected with Delta variants (21J: 33.3%, 21A: 33.3%), 1 with Omicron BA.1 (11.1%), 1 with recombinant (11.1%), and 1 with unknown variant (11.1%). In the 1,000 mg cohort group, 4 patients were infected with Omicron BA.1 (44.4%), 4 with recombinant (44.4%) and 1 with unknown variant (11.1%, tested positive on Day −1 but negative on Day 1). The clinical status of the patients was evaluated using an 8-point ordinal scal. At Day 1, all patients had a baseline clinical status assessed as score 4 (hospitalized, not requiring supplemental oxygen, but requiring ongoing medical care [SARS-CoV-2–related or other medical conditions]) or score 5 (hospitalized, requiring any noninvasive supplemental oxygen treatment). The mean clinical status score was 4.3 points for 250 mg cohort, 4.0 points for 500 and 1,000 mg cohorts. In the under 60 age group, the overall mean clinical status score was 4.3 points, compared to 4.0 points in the ≥ 60 age group.

## Safety

For both studies, safety was evaluated as primary endpoint, and independent data monitoring committees (IDMC) were set up. None of the stopping criteria were met.

**Phase 1a Study EXEVIR0102 (Healthy volunteers).** Overall, 22 of 30 (73.3%) volunteers in the Safety Set reported a total of 53 TEAEs. The relative frequency of volunteers with at least 1 TEAE was comparable between the cohorts (Table 2).

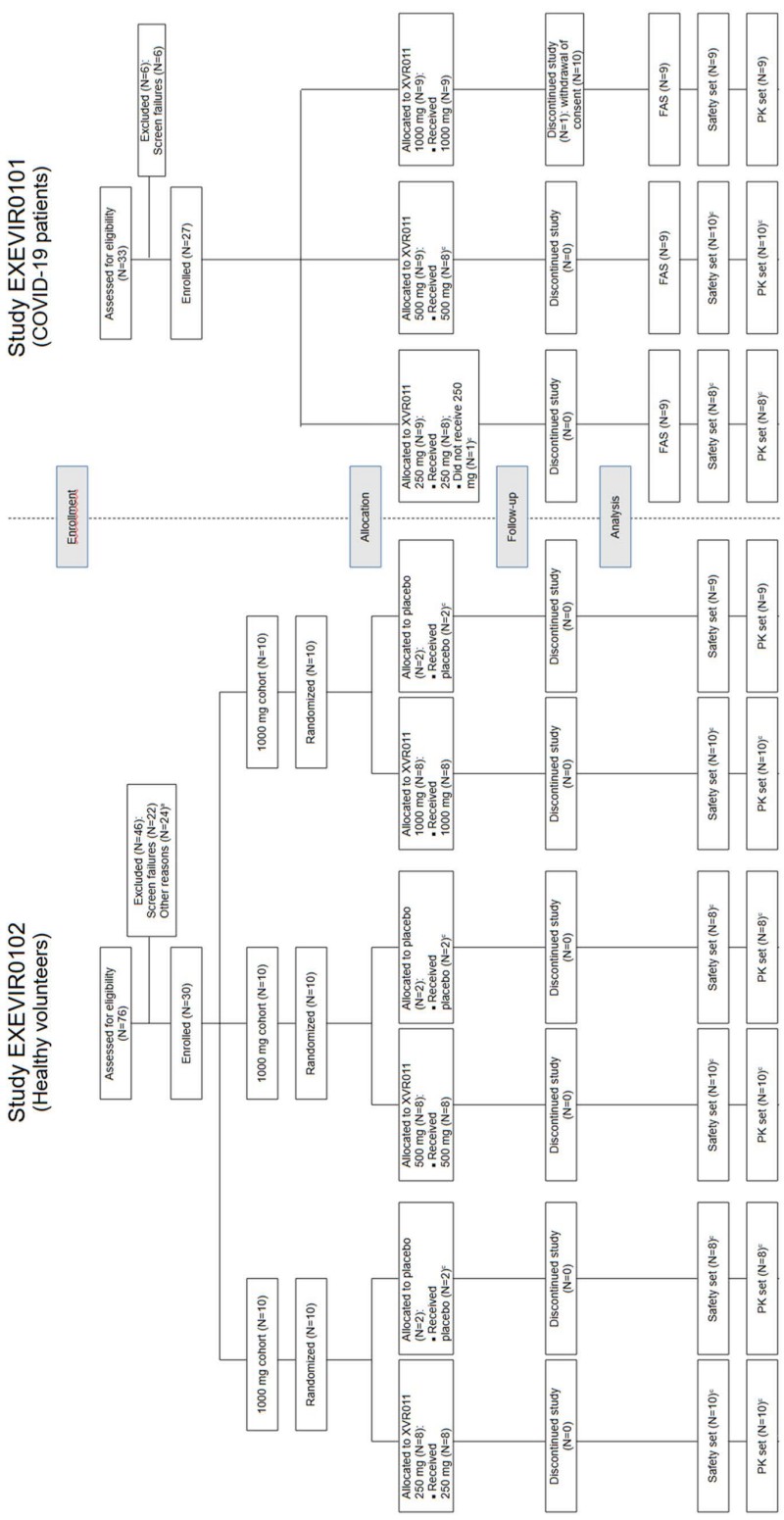

**Fig 1. Participant Disposition.** FAS: full analysis set (*I.e.* intention-to-treat (ITT)). Safety set: modified ITT, with dosing errors assigned to the dose actually received. PK set: modified ITT, safety set for which at least one time point has measurable XVR011.

**Table 2. Summary of adverse events (Safety set).**

| Category | EXEVIR0102 – Healthy volunteers | | | | | | | | | | EXEVIR0101 – Hospitalized COVID-19 patients | | | | | | | |
| | Placebo (N=6) | | Cohort 1 250 mg (N=8) | | Cohort 2 500 mg (N=8) | | Cohort 3 1,000 mg (N=8) | | Overall (N=30) | | Cohort 1 250 mg (N=8)[a] | | Cohort 2 500 mg (N=10)[a] | | Cohort 3 1,000 mg (N=9) | | Overall (N=27) | |
| | n (%) | E | n (%) | E | n (%) | E | n (%) | E | n (%) | E | n (%) | E | n (%) | E | n (%) | E | n (%) | E |
|---|---|---|---|---|---|---|---|---|---|---|---|---|---|---|---|---|---|---|
| TEAE | 5 (83.3) | 7 | 6 (75.0) | 20 | 5 (62.5) | 12 | 6 (75.0) | 14 | 22 (73.3) | 53 | 2 (25.0) | 3 | 3 (30.0) | 9 | 2 (22.2) | 3 | 7 (25.9) | 15 |
| TEAE of Grade ≥3 | 0 | 0 | 0 | 0 | 0 | 0 | 0 | 0 | 0 | 0 | 1 (12.5) | 1 | 1 (10.0) | 1 | 1 (11.1) | 1 | 3 (11.1) | 3 |
| Serious TEAE | 0 | 0 | 0 | 0 | 0 | 0 | 0 | 0 | 0 | 0 | 1 (12.5) | 1 | 1 (10.0) | 1 | 0 | 0 | 2 (7.4) | 2 |
| XVR011-related TEAE | 0 | 0 | 0 | 0 | 0 | 0 | 0 | 0 | 0 | 0 | 0 | 0 | 0 | 0 | 1 (11.1) | 2 | 1 (3.7) | 2 |
| XVR011-related TEAE of Grade ≥3 | 0 | 0 | 0 | 0 | 0 | 0 | 0 | 0 | 0 | 0 | 0 | 0 | 0 | 0 | 1 (11.1) | 1 | 1 (3.7) | 1 |
| XVR011-related serious TEAE | 0 | 0 | 0 | 0 | 0 | 0 | 0 | 0 | 0 | 0 | 0 | 0 | 0 | 0 | 0 | 0 | 0 | 0 |
| AE leading to discontinuation | 0 | 0 | 0 | 0 | 0 | 0 | 0 | 0 | 0 | 0 | 0 | 0 | 0 | 0 | 0 | 0 | 0 | 0 |
| AE leading to hospitalisation | 0 | 0 | 0 | 0 | 0 | 0 | 0 | 0 | 0 | 0 | 1 (12.5) | 1 | 1 (10.0) | 1 | 0 | 0 | 2 (7.4) | 2 |

AE = adverse event; E = number of events; N = number of participants in the analysis set; n = number of participants who experienced the event; TEAE = treatment-emergent adverse event.

[a]One participant in the 250-mg cohort erroneously received a 500-mg dose.

The most frequently reported TEAE by preferred term was headache, reported in 8 volunteers (26.7%) in various treatment groups. 3 volunteers (10.0%) reported each catheter site bruise, infusion site bruising, and nasopharyngitis. 2 volunteers (6.7%) reported each catheter site swelling and oropharyngeal pain. All other TEAEs were each reported by only 1 volunteer (3.3%). 1 volunteer in the 250-mg dose group reported four TEAEs (1 event of procedural pain, 1 event of gastroenteritis viral, and 2 events of nasopharyngitis), of moderate severity (Grade 2). All other TEAEs were of mild severity (Grade 1). No deaths, serious TEAEs, suspected unexpected serious adverse reactions, hypersensitivity reactions or severe (Grade ≥3) TEAEs were reported; no volunteers discontinued from the study due to a TEAE. None of the TEAEs was assessed as related to XVR011 and all TEAEs were resolved or disappeared by the end of the study, except for 1 event of nasopharyngitis in a participant in the placebo group for which the outcome was unknown. No clinically relevant findings were observed for clinical laboratory parameters, vital signs, electrocardiogram parameters, physical examination, concomitant medication, and local tolerability (infusion-related reactions).

**Phase 1b study EXEVIR0101 (Hospitalized COVID-19 patients).** Overall, 7 of 27 (25.9%) patients in the Safety Set reported a total of 15 TEAEs. Although more TEAEs were reported in the 500 mg cohort (9 events) compared with the other 2 cohorts (3 events each), the relative frequency of patients with at least 1 TEAE was comparable between the cohorts (Table 2). No obvious dose-response AE pattern was noted. No individual TEAE was reported in more than one patient (Table C in S1 Appendix). The majority of TEAEs were mild (Grade 1) or moderate (Grade 2) in severity. All TEAEs were resolved or disappeared by the end of the study.

No deaths or treatment-related serious TEAEs were reported. Two patients, 1 each in the 250- and 500 mg cohorts, experienced serious TEAEs (Grade 4 pulmonary embolism and Grade 3 chronic obstructive pulmonary disease, respectively) that lead to hospitalization. The events were assessed as not treatment-related. No patients discontinued from the study due to a TEAE. No infusion site reactions or hypersensitivity reactions were reported.

Variations in transaminase levels were observed within and across cohorts (Table C in S1 Appendix). Mean alanine aminotransferase (ALT) showed a post-treatment trend for dose-related increase across cohorts. However, peak variations from baseline were reached at different timepoints in the 3 cohorts. Three cases of clinically significant transaminase elevations were reported as TEAEs in 2 patients receiving the highest XVR011 dose. The first patient experienced a Grade 1 ALT elevation and a slight aspartate aminotransferase (AST) elevation on Day 6. One Grade 1 TEAE of

transaminases increase was reported, which was assessed as not treatment-related. The event was considered resolved by Day 16. The second participant experienced an ALT elevation on Day 3, which further increased on Day 8 to a value > 5x the upper limit of normal (ULN). A Grade 3 TEAE of ALT increase was described, which was assessed as treatment related. The event was considered resolved by Day 29. The same participant experienced an AST elevation on Day 4, which also further increased on Day 8 but remained < 2x ULN. A Grade 1 treatment related TEAE of AST increase was noted, which was considered resolved by Day 14. There was no indication of liver damage in any of the patients.

Based on laboratory results, a patient in the 500 mg cohort reported a Grade 2 TEAE elevation of serum cytokines on Day 14. The participant did not present other symptoms than general moderate fatigue and mild headache. The participant received medication and the event was considered resolved by Day 21 and assessed as not related to XVR011. The time interval between the XVR011 dose and the TEAE (about 2 weeks) suggests that the event was more likely related to COVID-19 than to XVR011.

The investigators considered abnormalities for all other clinical laboratory parameters, vital signs, electrocardiogram (ECG) parameters, and physical examination not clinically significant. The few clinically meaningful findings were generally considered unrelated to XVR011 and had been resolved by the end of the study.

## Pharmacokinetics

**Phase 1a study EXEVIR0102 (Healthy volunteers).** Following administration of single IV doses of 250, 500 and 1,000 mg XVR011, maximum XVR011 serum concentrations were reached shortly after the end of the infusion, at a median time to maximum serum concentration ($t_{max}$) ranging between 1.77 hours and 2.00 hours after the start of the infusion. Peak exposure ($C_{max}$) and systemic exposure ($AUC_{0-t}$ and $AUC_{0-inf}$) of XVR011 in serum increased with increasing doses (~2-fold increase between 2 consecutive cohorts for both parameters) (Table 3, Table F in S1 Appendix). Statistical analysis of dose-proportionality indicated that there was no significant deviation from dose-proportionality for $C_{max}$ and AUC ($p$-values were all > 0.05) (Table 3, Table F in S1 Appendix). The PK profiles of XVR011 were characterized by a geometric mean clearance (CL) ranging between 13.7 to 14.8 mL/h and a geometric mean half-life ($t_{1/2}$) ranging between 15.4 and 17.0 days (Table 3 and Fig 2A).

**Phase 1b study EXEVIR0101 (COVID-19 patients).** Following administration of single IV doses of 250, 500 and 1,000 mg XVR011, maximum XVR011 serum concentrations were reached shortly after termination of the infusion, at a median $t_{max}$ of less than 1 hour (0.833 h) in the 250 mg cohort, 1.317 hours post-dose in the 500 mg cohort, and more than 2 hours (2.167 h) post-dose in the 1,000 mg cohort. The differences in median $t_{max}$ between the cohorts reflect the differences in duration of the infusion. After the end of the infusion, XVR011 serum concentrations decreased in an approximately bi-exponential manner (Fig 2). Peak exposure ($C_{max}$) and systemic exposure ($AUC_{0-t}$ and $AUC_{0-inf}$) of XVR011 in serum increased with increasing doses, indicating that the serum PK parameters were approximately dose-proportional (no statistical analysis done). Limited data were available to estimate $t_{1/2}$, and geometric mean $t_{1/2}$ ranged from 11.9 to 13.7 days across cohorts (Table 3 and Fig 2A).

**Population PK model.** The XVR011 concentration data from Phase 1a and 1b studies appeared well described by a 2-compartment population PK model (Fig 2B, Figs C – E in S1 Appendix, Table E in S1 Appendix). Inter-individual variability was modelled using exponential parametrisations and was applied to the following parameters: CL, volume of the central compartment (Vc), volume of the peripheral compartment (Vp), and inter-compartmental clearance (Q) (Table 4). CL was estimated at 16.4 mL/h, Vc was estimated at 3,620 mL, and Vp at 4,730 mL, corresponding to a terminal $t_{1/2}$ of 15.9 days. Inter-individual variability appeared modest for all parameters (20.4% to 31.1%). Separate residual error terms were estimated for the two studies, with a higher residual error in Phase 1b study (13.7%) compared with Phase 1a study (9.8%). No other significant differences were found between the two studies/populations Fig 2B and Fig 4). Observed PK parameters from these Phase 1 trials appeared in line with the ones projected based on preclinical studies (Table D in S1 Appendix).

**Table 3. Summary of XVR011 pharmacokinetic parameters (PK Set).**

| Parameter | EXEVIR0102 – Healthy volunteers | | |
|---|---|---|---|
| | Cohort 1<br>250 mg XVR011<br>($N=8$) | Cohort 2<br>500 mg XVR011<br>($N=8$) | Cohort 3<br>1,000 mg XVR011<br>($N=8$) |
| $C_{max}$ (µg/mL) | 71.8 (20.1) | 155 (21.1) | 306 (28.2) |
| $t_{max}$ (h) | 2.00 (1.52; 2.08) | 2.00 (1.50; 2.00) | 1.77 (1.52; 2.00) |
| $AUC_{0-t}$ (µg.h/mL) | 16,639 (17.3) | 32,930 (24.1) | 71,281 (17.1) |
| $AUC_{0-inf}$ (µg.h/mL) | 17,124 (16.8) | 33,700 (24.7) | 72,973 (17.1) |
| $t_{1/2}$ (h) | 407 (9.3) | 369 (12.8) | 385 (10.1) |
| CL (mL/h) | 14.6 (16.8) | 14.8 (24.7) | 13.7 (17.1) |
| $V_z$ (mL) | 8,570 (23.9) | 7,898 (17.8) | 7,615 (18.1) |
| Parameter | EXEVIR0101 – Hospitalized COVID-19 patients | | |
| | Cohort 1<br>250 mg XVR011<br>($N=8$)[a] | Cohort 2<br>500 mg XVR011<br>($N=10$)[a] | Cohort 3<br>1,000 mg XVR011<br>($N=9$) |
| $C_{max}$ (µg/mL) | 64.25 (25.5) | $132.9^{N=9}$ (36.1) | 243.6 (27.0) |
| $t_{max}$ (h)[a] | 0.833 (0.68; 0.95) | $1.317^{N=9}$ (1.27; 1.38) | 2.167 (2.02; 22.88) |
| $AUC_{0-t}$ (µg.h/mL) | 10,320 (22.4) | $20030^{N=9}$ (17.7) | 42,810 (25.2) |
| $AUC_{0-inf}$ (µg.h/mL) | $10690^{N=2}$ (21.3) | $22380^{N=1}$ (NC) | $65200^{N=3}$ (16.7) |
| $t_{1/2}$ (h) | $312.6^{N=2}$ (19.5) | $284.9^{N=1}$ (NC) | $328.5^{N=3}$ (16.0) |
| %AUCex (%) | $20.24^{N=2}$ (41.4) | $17.87^{N=1}$ (NC) | $23.13^{N=3}$ (26.3) |

$AUC_{0-inf}$ = area under the serum concentration-time curve from time 0 to infinity; $AUC_{0-t}$ = area under the serum concentration-time curve from time 0 to time of last quantifiable concentration; %AUCex = percentage of $AUC_{0-inf}$ that is due to extrapolation beyond $t_{last}$; CL = clearance; $C_{max}$ = maximum serum concentration; CV% = coefficient of variation; N = number of participants included in the analysis set; $t_{1/2}$ = terminal phase half-life; $t_{max}$ = time to maximum serum concentration; $V_z$ = terminal phase distribution volume.

All participants included in the analysis set were included in the parameter calculation, unless indicated otherwise. Data represent geometric means (%CV), except for $t_{max}$ for which medians (range) are presented.

[a]One participant in the 250 mg cohort erroneously received a 500 mg dose.

## Pharmacodynamics

In the Phase 1b trial, the antiviral efficacy of a single IV infusion of XVR011 was evaluated in patients hospitalised for COVID-19 as a secondary endpoint. It should be noted that the study was not powered to detect statistically significant differences from baseline and across cohorts.

Viral load decreased in all cohorts with the same kinetics from baseline. At baseline, all patients had a clinical status score of 4 (hospitalised, not requiring supplemental oxygen, but requiring ongoing medical care [SARS-CoV-2–related or other medical conditions]) or 5 (hospitalised, requiring any supplemental oxygen). No mortality was observed. None of the patients required intensive care unit (ICU) admission nor mechanical ventilation. 3 patients in the 250 mg cohort (3/9, 33.3%) used noninvasive oxygen supplementation. Clinical score evolved similarly between all dose levels. Visualisation of the following parameters suggested a possible dose-response trend:

1] Time to COVID-19 related symptoms disappearance (TTSD) was on average 27.2, 25.7, and 20.3 days in the 250, 500 and 1,000 mg cohorts, respectively (Fig 3A).

2] The median time to hospital discharge (TTHD) was 14.8, 11.6, and 9.5 days in the 250, 500 and 1,000 mg cohorts, respectively (Fig 3B and Table G in S1 Appendix).

Percentage of patients having received a prior COVID-19 vaccine, number and type of prior COVID-19 vaccinations, as well as date since last vaccination, were similar in the 500 and 1,000 mg cohorts, while there were fewer

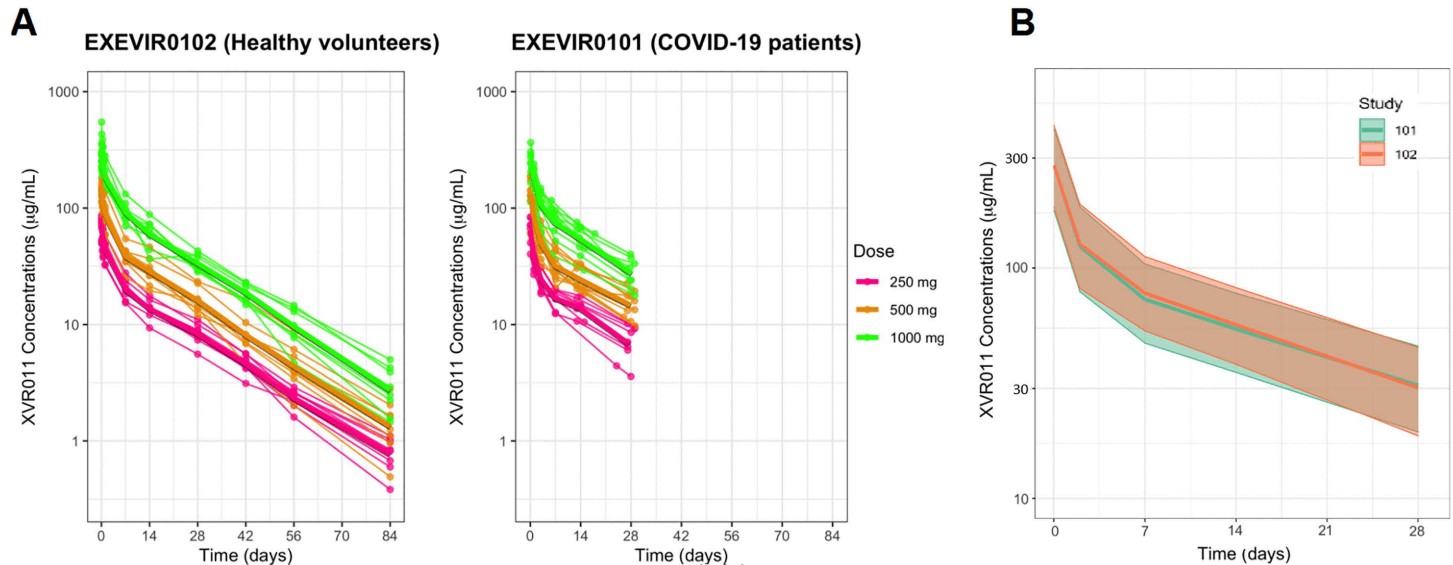

**Fig 2. Pharmacokinetics results (PK Set). A.** XVR011 serum concentration-time profiles in healthy volunteers (study EXEVIR0102) and hospitalized COVID-19 patients (study EXEVIR0101). Thin lines and points show individual data. Fat lines show geometric mean concentrations; **B.** XVR011 population PK in Phases 1a vs. 1b studies. Median and $CI_{95}$ are displayed.

**Table 4. Population PK Parameters for XVR011 Data From Studies EXEVIR0101 (healthy volunteers) and EXEVIR0102 (hospitalized COVID-19 patients) (PK Set).**

| Parameter (Unit) | Estimate | SE (%) | IIV (%) |
|---|---|---|---|
| CL (mL/h) | 16.4 | 3.1 | 21.9 |
| $V_c$ (mL) | 3,620 | 3.2 | 20.4 |
| $V_p$ (mL) | 4,730 | 4.1 | 24.2 |
| Q (mL/h) | 70.0 | 6.4 | 31.1 |
| Residual error EXEVIR0101 (%) | 13.7 | 14 | |
| Residual error EXEVIR0102 (%) | 9.77 | 9.2 | |

CL = clearance; IIV = inter-individual variability; Q = inter-compartmental clearance; SE = relative standard error for parameter estimate; $V_c$ = volume of central compartment; $V_p$ = volume of peripheral compartment.

vaccinees in the 250 mg cohort (Table H in S1 Appendix). However, the small study size, partially unknown history of prior COVID-19 infections, evolving standard of care, SARS-CoV-2 variant evolving from Delta in the 250 mg cohort to the less pathogenic Omicron BA.1 in the 1,000 mg cohort [31–33], and evolving potency of XVR011 between the Delta and Omicron BA.1 variants remain confounding factors, and the results should be interpreted with great care.

## Immunogenicity

The immunogenicity of one XVR011 dose in healthy volunteers and COVID-19 patients was evaluated as an exploratory objective in Phase 1a and 1b studies.

**Phase 1a study EXEVIR0102 (Healthy Volunteers).** Three out of 24 (12.5%) volunteers in the FAS who received XVR011 had ADA-positive samples (Table 5). A total of 7 samples were confirmed positive for ADAs against XVR011, all

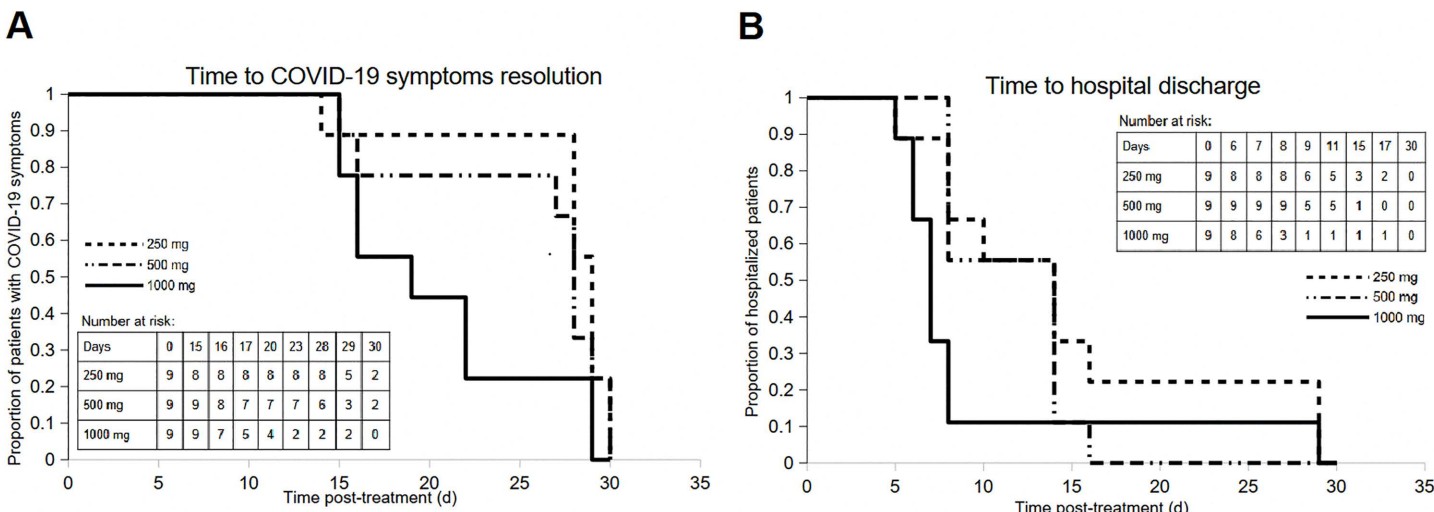

**Fig 3. Efficacy results (FAS). A.** Kaplan-Meier Curve of Time (days) to Disappearance of COVID-19 related symptoms (TTSD); **B.** Kaplan-Meier Curve of Time (days) to Hospital Discharge (TTHD). d: days.

**Table 5. Summary of ADA prevalence in the XVR011 phase 1a and 1b studies (FAS). ADA responses were confirmed and their titre was measured.**

|  | EXEVIR0102 – Healthy volunteers *n/N* (%) | EXEVIR0101 – Hospital-ized patients *n/N* (%) |
|---|---|---|
| **Pre-XVR011 dosing** | 3/24 (12,5%) | 1/27 (3,7%) |
| Cohort 1–250 mg | 0/8 | 0/9* |
| Cohort 2–500 mg | 1/8 (12,5%) | 1/9 (11.1%)* |
| Cohort 3–1,000 mg | 2/8 (25%) | 0/9 |
| **Post-XVR011 dosing** | 0/24 | 8/27 (29,6%) |
| Cohort 1–250 mg | 0/8 | 1/9 (11,1%)* |
| Cohort 2–500 mg | 0/8 | 3/9 (33,3%)* |
| Cohort 3–1,000 mg | 0/8 | 4/9 (44,4%) |

N = number of participants; n = number of participants with ADA-positive samples

* One participant in the 250 mg cohort received a 500 mg dose by mistake.

of which had a measurable titre. In the 500 mg cohort, 1 participant had a positive pre-dose sample while in the 1,000 mg cohort, 2 volunteers had ADA-positive samples at all timepoints (*i.e.,* pre-dose, Day 29, Day 85), with titres decreasing after XVR011 administration. These observations imply that there was no treatment-induced ADA formation (Table 5). For the 3 volunteers with ADAs against XVR011, the ADAs did not impact the pharmacokinetics of XVR011 (Fig 4).

**Phase 1b study EXEVIR0101 (COVID-19 patients).** Nine out of 27 (33.3%) patients in the FAS who received XVR011 had ADA-positive samples (Tables 5 and 6). In the 500 mg cohort, 1 participant had ADA-positive samples at all timepoints (i.e., pre-dose, Day 15, Day 29) with titres decreasing after XVR011 dosing. For 8 patients (1, 3 and 4 in the 250, 500 and 1,000 mg cohorts, respectively), at least 1 of the 2 post-dose samples was positive, whereas the pre-dose sample was negative. This raised the possibility of treatment-induced ADA formation (Tables 5 and 6). While the trials were not powered to study this endpoint, the results suggested no impact on the pharmacokinetic or pharmacodynamic parameters of XVR011 for the 9 patients with ADAs against XVR011: no clear difference on viral load reduction from baseline, time

**Table 6. Immunogenicity results (FAS). A. Efficacy parameters in EXEVIR0101 hospitalized COVID-19 patients, based on their treatment-induced ADA status; B. Characteristics of treatment-induced ADA-positive vs. -negative individuals in EXEVIR0101 hospitalized COVID-19 patients. The number followed by the percentage in parentheses is displayed. ADA+, treatment-induced ADA-positive; ADA-, treatment-induced ADA-negative; T2D: type II diabetes; d: days; y: years.**

|  | ADA$^+$ (N=8) | ADA$^-$ (N=18) |
|---|---|---|
| **A.** |  |  |
| Time to mild COVID symptoms, mean (d) | 8.5 (driven by 2/8 participants) | 4.4 |
| Time to COVID-19 symptom resolution, mean (d) | 25 | 24 |
| Time to hospital discharge, mean (d) | 12.1 | 11.6 |
| Lowest SaO$_2$ (%) | 94.9 | 95.1 |
| Duration of oxygen supplementation (d) | 4.5 (1 participant) | 1.4 |
| **B.** |  |  |
| Dose, median (mg) | 719 | 528 |
| Female gender | 5 (63%) | 11 (61%) |
| Age, mean (y) | 64 | 54 |
| Age > 50 y | 7 (88%) | 13 (72%) |
| Hypertension, any grade | 6 (75%) | 9 (50%) |
| Varicose | 1 (13%) | 2 (11%) |
| Cardiovascular disease, any | 7 (88%) | 9 (50%) |
| Obesity, any grade | 2 (25%) | 5 (28%) |
| T2D, any grade | 0 (0%) | 2 (11%) |
| Allergic rhinitis, any grade | 0 (0%) | 2 (11%) |
| Anxiety, any grade | 0 (0%) | 1 (5%) |

ADA$^+$, treatment-induced ADA-positive; ADA−, treatment-induced ADA-negative; SaO$_2$: oxygen saturation of the blood; d, days, T2D, type II diabetes; y, year.

to COVID-19 symptom resolution, time to mild COVID symptoms, time to hospital discharge, lowest oxygen saturation in the blood or duration of oxygen supplementation was observed (Fig 4, Table 6 and Table I in S1 Appendix). A post-hoc analysis suggested that the treatment-induced ADA were short-lived, peaking at day 14 post-dose and having partially or entirely disappeared at day 28/29 post-dose (end of study, Fig 5). Finally, the patients developing treatment-induced ADA had similar underlying medical conditions compared to those who did not (Table 6 and Table J in S1 Appendix).

## Discussion

Here, two Phase 1 clinical studies were performed with the objective to evaluate the safety, tolerability, PK, efficacy, and immunogenicity of XVR011, a VHH-Fc construct targeting SARS-CoV-2, in healthy volunteers (Phase 1a) and in patients hospitalized with mild to moderate COVID-19 (Phase 1b).

Administration of single IV doses of 250, 500 and 1,000 mg XVR011 appeared safe and well tolerated in both healthy volunteers and hospitalized patients with mild to moderate COVID-19. In the Phase 1a study, 22 volunteers (73.3%) reported a total of 53 TEAEs, with none reported as being related to XVR011. In the Phase 1b study, 7 patients (25.9%) reported a total of 15 TEAEs, with one reported as being related to XVR011 in the highest dose group. No obvious dose-response AE pattern was noted and all TEAEs resolved or disappeared by the end of the study. In the Phase 1b study, variations in transaminase levels were observed within and across cohorts, with a trend for a higher mean ALT following treatment with a higher dose. Clinically significant transaminase elevations were reported as TEAEs in 2 patients receiving the XVR011 dose. As no signs of liver damage were reported in any of the patients, the clinical relevance of these findings may be limited. In comparison, the incidence of drug-related TEAE for the conventional anti-SARS-CoV-2

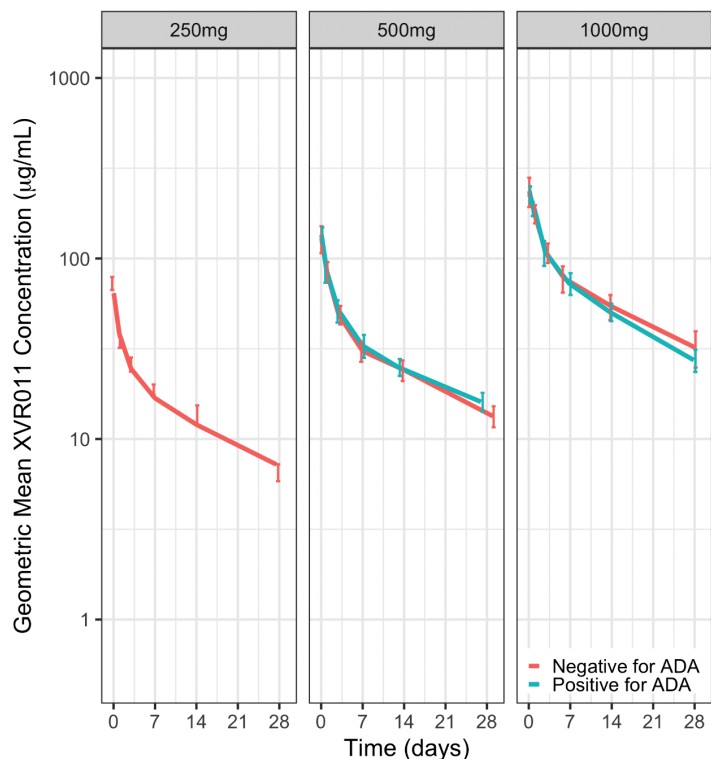

**Fig 4. Impact of anti-drug antibodies (ADA) on the XVR011 Serum Concentration-time Profiles in Healthy Volunteers (Study EXEVIR0102) and COVID-19 Patients (Study EXEVIR0101).**

therapeutic antibody bebtelovimab was 28.6% (150 mg), 25% (600 mg), 14.3% (1,200 mg), 50% (2,400 mg) and 37.5% (placebo) in healthy volunteers [34].

Pharmacokinetic analysis indicated overall low-to-moderate variability in the PK of XVR011 with a clear dose-response relationship for $C_{max}$ and AUC. The estimated half-life of XVR011 of 16 days was slightly shorter than the typical 21 days reported by Mankarious *and colleagues*, for naturally occurring antibodies [35]. It appeared, however, in line with the half-life of 14 days after a first dose of envafolimab, another VHH-Fc construct tested in patients with solid tumors [8,9]. It is important to note that the half-life of any antibody, either a single domain or a conventional IgG1 antibody, is highly variable and influenced by many factors, including Fc glycosylation pattern, binding affinity to FcRn, and interactions with the target antigen. In addition, individual patient variability and specific disease conditions may impact the PK of therapeutic antibodies [36,37]. When comparing the half-life of XVR011 with anti-SARS-CoV-2 monoclonal antibodies (without any Fc modifications to extend the half-life), the half-life of XVR011 fell within the range of these therapeutic antibodies, ranging from 11.5 days (for bebtelovimab) to 31 days (for casirivimab) [38,39]. Moreover, the results of XVR011 and the reported anti-SARS CoV-2 monoclonal antibodies appeared in line with those reported for therapeutic IgG antibodies, exhibiting a plasma half-life of 6–32 days in humans [40,41]. No significant difference in PK parameters between hospitalized patients and healthy volunteers was observed in the population PK analysis. Concentrations of XVR011 were more variable in COVID-19 hospitalized patients than in healthy volunteers. This may be linked to the presence of SARS-CoV-2 virus, the target of XVR011, at variable levels in the hospitalized COVID-19 patients (target-mediated drug disposition) and/or to their variable health status (underlying conditions and/or COVID-19 disease, responsible for higher distribution to the damaged airways).

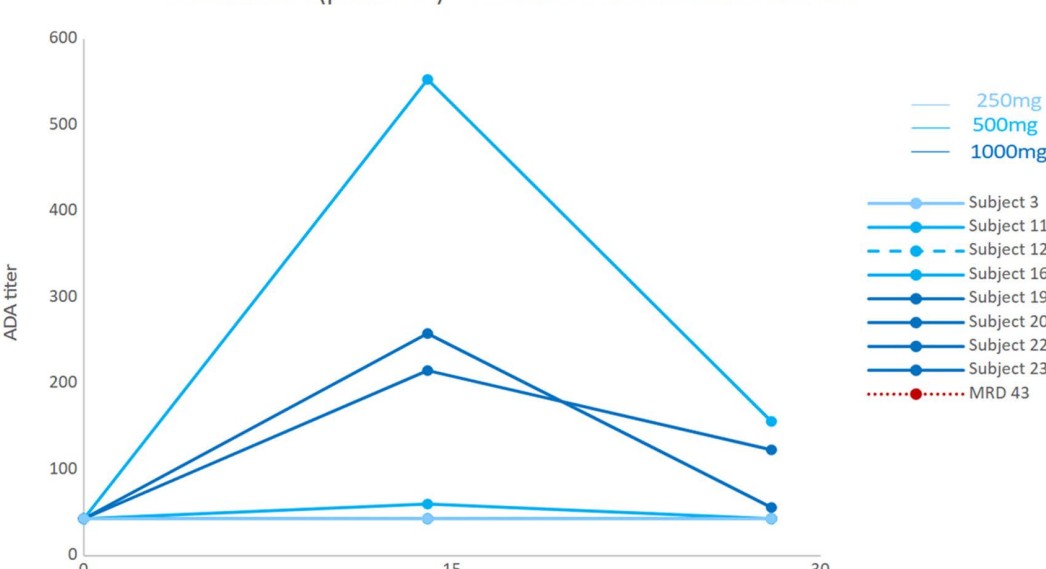

**Fig 5. Individual kinetics of treatment-induced anti-drug antibodies (ADA) responses in EXEVIR0101 hospitalized COVID-19 patients.** MRD: minimal required dilution.

   Anti-drug antibodies (ADA) may affect the pharmacokinetics, efficacy and/or safety of therapeutic antibodies. Hence, immunogenicity can be a critical challenge in the development of antibody therapeutics [42], irrespective of a fully human origin or, in this case, a humanized VHH fused to a human Fc. While humanization significantly reduces immunogenicity compared to earlier chimeric or animal-derived antibodies, it doesn't eliminate it entirely, as the antigen-binding complementarity-determining regions (CDRs) can still elicit an immune response [43,44]. However, ADA development against conventional human antibodies also varies widely and is dependent on numerous factors, including antibody structure, formulation and an individual's genetic predisposition, age, and disease status [45–48]. Here, immunogenicity was evaluated as an exploratory objective in Phase 1a/1b studies. Interestingly, in the Phase 1a study, none of the healthy volunteers showed treatment-induced ADA formation. In the Phase 1b study, ADAs were formed following treatment in ~31% of patients. By way of comparison, another VHH-Fc construct, envafolimab, induced ADA formation in ~36% of the solid tumor patients (no clinical trial was performed in healthy volunteers), with no impact on PK and PD [23,24]. Of the other two approved VHH-based products, caplacizumab (bivalent VHH) and ozoralizumab (trivalent VHH) induced 9–14.3 and 39.1% ADAs, respectively, in clinical trials [18–20]. Bebtelovimab, a conventional anti-SARS-CoV-2 antibody, induced 12.5% ADA responses in healthy volunteers [34]. Immunogenicity of conventional antibodies is highly variable, ranging from 0% to 96%, with an average of 12.3% [49]. VHH-Fc molecules such as XVR011 fall within this range. XVR011-induced ADA were present primarily in the mid and high dose groups of the COVID-19 hospitalized patient study and were short-lived. It is therefore possible that the highly pro-inflammatory health status of the hospitalized COVID-19 patients may have triggered a 'temporary' ADA response to high drug concentrations. Importantly, while ADAs can lead to the formation of immune complexes which can affect both PK and PD properties (hypersensitivity reactions, loss of exposure, and efficacy), there was no suggestion in our studies that the ADAs impacted the PK or PD of XVR011 [50].
   The studies presented here have limitations. Both studies had a limited number of participants, the Phase 1b was not designed to demonstrate statistical efficacy (small sample size, no placebo control), the history of prior COVID-19 infections

was unknown, the duration since last vaccination was variable and the patients were treated with different standard of care (SOC), infected with different variants of variable pathogenicity and varying potency of XVR011 against these variants. Hence, all these factors may impact interpretation of safety, PK, PD, and/or immunogenicity signals. Therefore, conclusions must be drawn carefully. Nonetheless, the studies highlight the potential of the VHH-Fc platform and support the design and further development of VHH-Fc antibody constructs for the prevention and/or treatment of COVID-19 and other infectious diseases.

## Supporting information

**S1 Appendix. Supplementary Information with additional text, tables, and figures.**
(PDF)

**S1 Checklist. CONSORT 2025 checklist.** Hopewell S, Chan AW, Collins GS, Hróbjartsson A, Moher D, Schulz KF, and colleagues. CONSORT 2025 Statement: updated guideline for reporting randomised trials. *BMJ*. 2025;388:e081123. https://dx.doi.org/10.1136/bmj-2024-081123. This checklist is licensed under the Creative Commons Attribution 4.0 International License (CC BY 4.0; https://creativecommons.org/licenses/by/4.0/).
(PDF)

**S2 Checklist. CONSORT-CONSERVE 2025 checklist.**
(PDF)

**S1 Protocol. Phase 1a EXEVIR0102 clinical study protocol (amendment 1).**
(PDF)

**S2 Protocol. Phase 1b EXEVIR0101 clinical study protocol (amendment 6).**
(PDF)

## Acknowledgments

We acknowledge the contribution of the volunteers and their families for participating in the clinical studies, as well as the healthcare workers who made the trial possible. We thank Bernard Miserez, and the agency Emtex, for medical writing support funded by Exevir.

## Author contributions

**Conceptualization:** Ellen Jansen, Viki Bockstal, Angélique Boerboom, Dominique Tersago.

**Data curation:** Ellen Jansen, Dominique Tersago.

**Formal analysis:** Ellen Jansen, Viki Bockstal, Florence Herschke, Per Olsson Gisleskog, Dominique Tersago.

**Funding acquisition:** Viki Bockstal, Angélique Boerboom, Dominique Tersago.

**Investigation:** Ellen Jansen, Angélique Boerboom, Salah Hadi, Natalia Gaibu, Michel Moutschen, Dominique Tersago.

**Methodology:** Ellen Jansen, Viki Bockstal, Manuela Rinaldi, Dominique Tersago.

**Project administration:** Ellen Jansen, Manuela Rinaldi, Angélique Boerboom.

**Supervision:** Viki Bockstal, Dominique Tersago.

**Visualization:** Florence Herschke, Per Olsson Gisleskog.

**Writing – original draft:** Viki Bockstal, Florence Herschke, Manuela Rinaldi.

**Writing – review & editing:** Ellen Jansen, Viki Bockstal, Florence Herschke, Per Olsson Gisleskog, Manuela Rinaldi, Angélique Boerboom, Salah Hadi, Natalia Gaibu, Michel Moutschen, Dominique Tersago.

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
