## [Editor Report · Decision Letter 0]

9 Jan 2025

Dear Dr Herschke,

Thank you for submitting your manuscript entitled "First-in-Human characterization of XVR011 (rimteravimab), the first VHH-Fc construct tested in randomized clinical trials, in healthy adults and patients hospitalized for mild-to-moderate COVID-19" for consideration by PLOS Medicine.

Your manuscript has now been evaluated by the PLOS Medicine editorial staff as well as by an academic editor with relevant expertise and I am writing to let you know that we would like to send your submission out for external peer review, pending consideration of the full protocol.

Please also note the following:

Please provide the IRB-approved protocol with your submission as supplementary information, identify in the Methods the IRBs that approved the study, and the Data Safety Monitoring Boards involved, or clarify the method of safety oversight. Please also state in the abstract the primary and secondary study objectives, and signpost all exploratory objectives throughout the manuscript. Please note that the link in reference 3 may not work. We further note that Part 2 of the Phase 1b trial is not included in the manuscript. Please clarify to the editors the status of Part 2, whether that study is complete and if so, whether the results are submitted for publication elsewhere, in view of the potential impact on conceptual novelty for this report. Please include a limitations section in the Discussion and discuss whether ADA responses have been observed in other trials of VHH-based therapeutics. Lastly, 4 VHH-based therapeutics have been approved by different regulatory agencies, therefore please amend the Introduction accordingly.

Please re-submit your manuscript within two working days, i.e. by Jan 13 2025 11:59PM. Please email the editor if you require longer.

Kind regards,

Alison Farrell, Ph.D.

Senior Editor

PLOS Medicine

---

## [Decision Letter · Decision Letter 1]

27 Aug 2025

Dear Dr Herschke,

Many thanks for submitting your manuscript "First-in-Human characterization of XVR011 (rimteravimab), the first VHH-Fc construct tested in randomized clinical trials in healthy adults and patients hospitalized for mild-to-moderate COVID-19" (PMEDICINE-D-25-00069R1) to PLOS Medicine. I apologize for the delay in conveying to you our decision on the study as we had difficulty securing sufficient reviewers. The paper has now been reviewed by 3 subject experts and a statistician; their comments are included below and can also be accessed here: ********

As you will see, reviewers 1, 3 and 4 find the study of potential interest. After discussing the paper with the editorial team and an academic editor with relevant expertise, I'm pleased to invite you to revise the paper in response to the reviewers' comments. We plan to send the revised paper to some or all of the original reviewers, and we cannot provide any guarantees at this stage regarding publication.

We ask that you address the comments of reviewers 1, 3 and 4 in full, in addition to the academic and staff editors' comments. Please ensure that the trial reporting follows CONSORT 2025 reporting and that claims are tempered and acknowledge the small study size and inherent limitations.

Please note: we ask that you clarify the delay in reporting the study findings.

We ask that you submit your revision by Sep 17 2025 11:59PM. However, if this deadline is not feasible, please contact me by email, and we can discuss a suitable alternative.

Don't hesitate to contact me directly with any questions (afarrell@plos.org).

Best regards,

Alison

Alison Farrell, Ph.D.

Senior Editor

PLOS Medicine

afarrell@plos.org

Comments from the academic editor:

The weaknesses of the study are mostly covered by the other reviewer's comments, including the small sample size of the Phase 1 a study, which limits conclusions on rare safety signals and efficacy. The Phase 1b was open-label, had no placebo group, was also underpowered, and conducted amid shifting variants and standards of care. Reported “dose-response trends” must be interpreted cautiously. XVR011 lost activity against Omicron BA.2, undermining long-term therapeutic relevance.

Although presented as clinically irrelevant, the relatively high ADA incidence in patients (~30%) raises concern for broader use, especially since immunogenicity is unpredictable in diverse populations.

Some of the authors are employees or contractors of ExeVir Bio, the sponsor — there may be some bias in interpretation.

The manuscript is long and somewhat repetitive, with extensive technical detail that might be better placed in supplementary material.

Comments from the editors:

* Please revise your title so that it complies with to PLOS Medicine's style. Your title must be nondeclarative and not a question. It should begin with main concept if possible. "Effect of" should be used only if causality can be inferred, i.e., for an RCT. Please place the study design ("A randomized controlled trial," "A retrospective study," "A modelling study," etc.) in the subtitle (ie, after a colon). Please remove the first in human from the title.

* Please review your text for claims of novelty or primacy (e.g. 'for the first time') and remove this language. In addition, please check that any use of statistical terms (such as trend or significant) are supported by the data, and if not please remove them.

Please clarify why there are no authors from the Netherlands.

Please acknowledge the healthcare workers who made the trial possible.

Please add the names of the ethics committee and regulatory boards that approved the trials and protocol (s).

Please clarify which boards approved any study amendments and ensure that the amendment(s) are detailed in the protocol, along with dates of amendments.

Please state in the manuscript the dates of first and last participant enrolment (first paragraph of the results section).

Identify the study sites in the Methods.

Please specify the name of the data safety monitoring board or its members’ location.

* Please confirm that your abstract complies with our requirements, including format. There are only three sections: Background, Methods and Findings, and Conclusion. Please provide all the information relevant to this study type https://journals.plos.org/plosmedicine/s/submission-guidelines#loc-abstract

In particular, please ensure that your Abstract is CONSORTed according to the CONSORT 2025 guidelines and provide a CONSORT 2025 checklist.

* If all secondary outcomes described in the protocol are included in the manuscript, please briefly describe them in the Abstract. If only a subset is reported in this study, please clarify why in the Methods and do not report in the Abstract.

The Background section of the Abstract must present information that explains the rationale for the study. Please do not begin by saying this is the first trial of a VHH-Fc in this population.

Please correct the grammatical errors in the Abstract and use the active voice.

Please state SARS-CoV-2 before Omicron BA.2.

Please remove discussion of trends in the Abstract.

Please avoid causal claims of causality in the Abstract where they may not be supported and temper your conclusions.

The last sentence of the section Methods and Findings must state the technological limitations of the study.

Please add the clinical trial identifiers after the Abstract.

Please clarify your statement in the Abstract to explain how it was determined that treatment emergent adverse events were not related to the treatment.

Line 86: add “to our knowledge”

Line 124: please clarify why the effects of the LALA mutation would be of benefit.

Please confirm that the trials were prospectively registered.

Please ensure that the trial numbers are correct and embed links in the Word document.

Please add more detail to the Introduction on conventional mAb trials for COVID-19.

Please add details on enrollment of patients: were these patients who would have been hospitalized or were they recruited and hospitalized based on the trial protocol?

Please include the patient questionnaire in the supplementary information.

Please avoid the use of ‘subjects’.

Please explain why there was a SARS-CoV-2-negative participant in the Phase 1b study.

In the introduction, please clarify if the VHH-Fc was shown to neutralize SARS-CoV-2 Delta.

With respect to dose-response trends, please explicitly state that the observed differences were not statistically significant.

Please include a short, non-technical Author Summary of your research to make findings accessible to a wide audience that includes both scientists and non-scientists. The Author Summary should immediately follow the Abstract in your revised manuscript. This text is subject to editorial change and should be distinct from the scientific abstract. Ideally each sub-heading should contain 2-3 single sentence, concise bullet points containing the most salient points from your study. In the final bullet point of ‘What Do These Findings Mean?’ Please include the main limitations of the study in non-technical language.

Please reduce the Discussion to 1000 words. Focus on the issues of next steps for VHH-Fc constructs, challenges for infection and for elicitation of ADA responses.

The Discussion must include a paragraph on study limitations.

Please adhere to the PLOS Data Availability guidelines.

Please avoid the use of both red and green in the same figures.

Please ensure Figure legends are included, for the main text and supplementary data.

Comments from the reviewers:

Reviewer #1: The research question is an important one to the community of researchers in this general area because it is the first publication of a VHH-Fc tested in randomized clinical trials. By providing a thorough analysis of the safety, tolerability, pharmacokinetics and immunogenicity of the VHH-Fc in two randomized clinical trials, the results of this publication provide a substantial advance over existing knowledge, with clear implications for patient care, public policy, or clinical research agendas.

While other VHH based therapies have been FDA approved and are in different stages of clinical trials this is the first report of a bivalent VHH-Fc construct being evaluated in humans for safety, tolerability, pharmacokinetics and immunogenicity. The construct evaluated was a bivalent VHH-Fc construct (two identical VHHs fused to an Fc silenced for effector function, using the LALA mutation. The VHH is a the previously characterized VHH72 isolated in llamas against SARS-CoV-1 and was demonstrated to bind to an epitope inaccessible to conventional human antibodies and has potent neutralization of SARS-CoV-1 and SARS-CoV-2. The neutralization is based on both direct competition with ACE2 biding to the spike RBD through steric hindrance and by trapping the RBD in the unstable up conformation resulting in shedding if the S1 subunit. While efficacy was evaluated as an exploratory objective as part of the phase 1b study, the study was not set-up detect statistically significant differences from baseline across cohorts and no placebo control was included for the treatment group. In addition, In March 2022, the study was prematurely terminated before Part 2 was initiated due to the loss of neutralization potency by XVR011 against the Omicron variant BA.2 that was spreading worldwide at the time. This investigation of the first VHH-Fc fusion construct in humans for tolerability, safety, pharmacokinetics, and immunogenicity represents a milestone for further development of VHH-Fc constructs as effective therapeutics for infectious disease especially multivalent constructs that are less likely to be susceptible to antigenic drift and antibody escape.

To improve the manuscript the authors should give more information about data from clinical trials for conventional anti-SARS-CoV-2 antibodies and compare the dose, tolerability, PK/PD, immunogenicity and ADA. The authors should also provide references and explain more about why ADA might be a greater concern with a llama derived VHH based construct in comparison to a humanized version or a conventional human antibody.

While in most cases the data and analyses fully support the claims, The exception is the statement that the construct is working strictly through neutralization. Several studies have shown that the LALA mutation does not completely abrogate all effector function. The authors should provide a reference for or results of effector function evaluation including ADCC, and CDC for this construct to confirm silencing.

minor issues:

Line 210 on page 9. Right after Figure S2 Error! Reference source not found.. has two periods.

Page 20 line 535 should be neutralization not neutralisation

Reviewer #2: This article offers two small studies of a VHH-FC construct. In the first study an escalating dose proved safe and well tolerated. The drug had expected PK behavior. While it was offered to a few subjects late in the course of cOVID-19 infection hospitalized, resistance of the Omicron stopped this exploration. In addition, other mAB studies (e.g. REGENCOV) demonstrated lack of benefit of antibodies in hospitalized patients. The data are several years old. While the authors end the paper arguing that these results are a milestone in VHH construct biology, it is hard to see how the results lead to this conclusion?

Reviewer #3: In this study, Jansen et al. ran two phase 1 trials testing the safety of a SARS-CoV-2 neutralising VHH-Fc construct in healthy adults and COVID-19 hospitalised patients. They found that the VHH-Fc drug (XVR011) was well tolerated with mild to moderate treatment related adverse events. Anti-drug antibody was detected in ~31% of COVID-19 patients but not in the healthy adults, though ADA was not reported to affect PK or PD of the drug. While there was a dose-response trend to time of resolution of symptoms or hospital discharge with XVR011 treatment, there are quite a few confounding factors that may cause this (described more in comments below). Overall this was a well described clinical study with detailed methods and results.

Major comments:

For the phase 1b study, the dose response trend for the time of resolution of COVID symptoms or time to hospital discharge may have been confounded by other factors including number of previous vaccinations or infections. If available, the authors should report the number of previous vaccinations and infections and type of prior vaccination in a table. It is important to know whether these factors were well matched between the different groups.

Also, the 1000mg group had more Omicron and recombinant infections than the 250 and 500mg groups, which mostly had Delta infection. Since Omicron and recombinant variants may have had different infection kinetics and outcomes compared to Delta, the time to COVID-19 symptom resolution or time to hospital discharge could have been due to the infecting variant, rather than treatment with XVR011. As such, the authors should tone down the fact that there was a dose-response trend for treatment throughout the article and abstract. Especially given there was no statistical basis for these statements.

It was interesting that there was only detectable ADA in the COVID patients and not in the healthy controls. Do the authors know whether the ADA were directed towards the llama VHH portion (VHH72) rather than the human IgG-LALA portion? Can the authors run a similar ADA assay with VHH72 alone and with an IgG-LALA control antibody of another specificity?

Minor comments:

Is reference 42 a full reference?

Figure 2 - the mean lines would be clearer if you could make the thin lines more faint or transparent.

Reviewer #4: Statistical review

This paper reports two phase I trials of XVR011 in healthy volunteers and mild-moderate covid patients. As is appropriate for phase I trials, much of the results was not subject to hypothesis testing. I had some minor comments on the statistical aspects of the paper:

1. Abstract: "Efficacy in subjects hospitalized with mild to moderate COVID-19 was evaluated as a secondary objective in the Phase 1b study." - it would be useful to add the endpoint used to measure efficacy here.

2. Abstract: "Although the study was not powered to detect statistically significant differences, a dose-response trend was observed for time to disappearance of COVID-19-related symptoms, as well as for time to hospital discharge." - I did not see that this statement was supported by the results as currently written as it initially made me assume some statistical testing had been done. Perhaps if it were rewritten along the lines of 'Visualisations of time to… suggested a possible dose-response trend', this might be clearer to the reader.

3. Line 176-177: I am not convinced that simple randomisation would be the approach used here as would mean there's a decent chance that (for example) 2 sentinel subjects would get placebo or dose. I suspect it was a block randomisation approach that ensured the number getting placebo/drug was as required?

4. Line 288-289: I am not familiar with the power model with mixed effects, can a bit more detail about what the model is estimating, and/or a reference, be added?

5. Lines 385-386: is Figure S5 the correct reference for this? I did not follow how it connected to the statement. It would be interesting to know what the estimated power parameter was, together with a confidence interval - is this available?

6. Line 498: I would recommend this is rewritten similarly to my comment 2.

James Wason

Any attachments provided with reviews can be seen via the following link: ********

---

* Please upload any figures associated with your paper as individual TIF or EPS files with 300dpi resolution at resubmission; please read our figure guidelines for more information on our requirements: http://journals.plos.org/plosmedicine/s/figures. While revising your submission, we strongly recommend that you use PLOS's NAAS tool (https://ngplosjournals.pagemajik.ai/artanalysis) to test your figure files. NAAS can convert your figure files to the TIFF file type and meet basic requirements (such as print size, resolution), or provide you with a report on issues that do not meet our requirements and that NAAS cannot fix.

After uploading your figures to PLOS's NAAS tool - https://ngplosjournals.pagemajik.ai/artanalysis, NAAS will process the files provided and display the results in the "Uploaded Files" section of the page as the processing is complete.

If the uploaded figures meet our requirements (or NAAS is able to fix the files to meet our requirements), the figure will be marked as "fixed" above. If NAAS is unable to fix the files, a red "failed" label will appear above.

When NAAS has confirmed that the figure files meet our requirements, please download the file via the download option, and include these NAAS processed figure files when submitting your revised manuscript.

* Please ensure that the study is reported according to the CONSORT 2025 guideline and include the completed CONSORT 2025 checklist as Supporting Information. When completing the checklist, please use section and paragraph numbers, rather than page numbers. Please add the following statement, or similar, to the Methods: "This study is reported as per CONSORT 2025 guideline (S1 Checklist)."

FIGURES AND TABLES

SUPPLEMENTARY MATERIAL

REFERENCES

RCTs:

* PLOS Medicine requires that all trials be prospectively registered in one of registries recognized by WHO. Please ensure that study registration details are included in the Methods section.

* Please structure the Methods section using the following sub-headings: Study design and participants, Randomization and masking, Procedures, Outcomes, Statistical analysis.

* The following outcomes measures [ADD DETAILS AS NEEDED OR DELETE BULLET POINT] appear to differ between the submitted manuscript and the protocol [and/or trial registry]. Please clarify and explain all discrepancies between the paper and protocol. If the outcomes were not prespecified in the protocol, please define them in the Methods (Outcomes section) as post hoc and explain why they were added. Post-hoc comparisons should be presented as hypothesis generating rather than conclusive.

* Please ensure that all prespecified outcomes (primary, secondary, and exploratory) are listed in the Methods/Outcomes section and indicate whether there are outcomes that are not presented in the current report.

* Please specify the dates (Month Day, Year) during which study enrollment and follow up occurred.

* Please include absolute numbers wherever you report percentages; eg, n/N (%)

* Please present the safety data for the study including numbers of specific events and whether or not adverse events are thought to be related to treatment. AEs should be reported in the abstract, per CONSORT and CONSORT-Harms.

* Please complete the CONSORT checklist (https://www.equator-network.org/reporting-guidelines/consort/) and ensure that all components of CONSORT are present in the manuscript, including how randomization was performed, allocation concealment, blinding of intervention, definition of lost to follow-up, power statement. When completing the checklist, please use section and paragraph numbers, rather than page numbers.

* Please report your abstract according to CONSORT for abstracts, following the PLOS Medicine abstract structure (Background, Methods and Findings, Conclusions) https://www.equator-network.org/reporting-guidelines/consort-abstracts/

* If your trial had to undergo important modifications in response to extenuating circumstances, please complete the CONSERVE-CONSORT checklist and provide in your Supporting Information; (https://www.equator-network.org/reporting-guidelines/guidelines-for-reporting-trial-protocols-and-completed-trials-modified-due-to-the-covid-19-pandemic-and-other-extenuating-circumstances-the-conserve-2021-statement/). When completing the checklist, please use section and paragraph numbers, rather than page numbers.

* In keeping with our commitment to Open Science, please include the study protocol document and analysis plan (including any amendments) as Supporting Information to be published with the manuscript if accepted.

* Please note that PLOS Medicine requires prospective, public registration of a data sharing plan (as part of mandatory clinical trials registration) for all clinical trials that began enrollment on or after January 1, 2019, in accordance with ICMJE requirements.

---

## [Decision Letter · Decision Letter 2]

20 Feb 2026

Dear Dr. Herschke,

Thank you very much for re-submitting your manuscript "First-in-Human characterization of VHH-Fc construct XVR011 (rimteravimab) in healthy adults and patients hospitalized for mild-to-moderate COVID-19: two Phase 1 randomized clinical trials." (PMEDICINE-D-25-00069R2) for review by PLOS Medicine.

I have discussed the paper with my colleagues and the academic editor and it was also seen again by 3 reviewers. I am pleased to say that provided the remaining editorial and production issues are dealt with we are planning to accept the paper for publication in the journal.

[LINK]

We look forward to receiving the revised manuscript by Feb 27 2026 11:59PM.

Sincerely,

Alison Farrell, Ph.D.

Senior Editor

PLOS Medicine

plosmedicine.org

Requests from Editors:

Use the active voice throughout.

**Please note the requirements to remove claims of primacy in the General Formatting comments below. This needs to be removed throughout the manuscript and from the title.

Please keep the tense consistent in the Abstract, e.g. line 78, switch to past tense.

Please provide additional background context to the Background section of the Abstract. The last sentence should explain the problem tackled in the manuscript.

Secondary objectives can be presented in the Abstract if ALL secondary objectives are reported in the manuscript. If only some are reported in the manuscript, this must be disclosed, and they must be removed from the Abstract. At present the Abstract lacks sufficient detail about how efficacy was assessed.

Line 78: add ‘of XVR011’ after tolerability.

Please indicate length of follow up of treated hospitalized patients in the Abstract.

Please indicate that descriptive statistics were used.

Please revise your financial disclosure to acknowledge that Exevir is a funder given that some of the authors were contracted by Exevir to perform the work. It does not seem correct to indicate that the finders had no role in study design, data collection and analysis, decision to publish, or preparation of the manuscript.

Author Summary—revise the title please

Please remove the specifics of antibody halflife from the Author Summary

Introduction: please explain why XVR011 was developed, and targeting which SARS-CoV-2 variant.

Line 156: Complimentary should be Complementary

Line 159: add ‘individuals’ after immunocompromised.

The last sentence of the Introduction is repeated twice. Please delete one sentence and add a clarifying sentence to explain what the two studies aimed to investigate with respect to XVR011

Please reorder the Introduction to start with the second paragraph. The first paragraph should be moved later.

Methods:

Please include the exact participant enrolment dates for the participants of EXEVIR0102 in the first paragraph.

Please explain how participants were recruited and if compensated.

Please include information on participant consent for the two trial in their initial description in the Methods.

The Methods need to include a specific section describing the protocol amendments.

**The manuscript does not mention a placebo control for the Phase 1a in healthy volunteers, it is only evident from Table 2. This needs to be clarified from the outset, including in the Abstract.

Results:

Line 318: should primate be plural?

Line 338: please fix spacing of “6patients”

Line 447: revise statement as per reviewer comments.

Line 450: replace ‘less’ with ‘fewer’

Discussion:

Line 518: replace “can, but not necessarily” with “may” (phrasing as written is awkward)

Please temper statements, e.g. ‘milestone’, It is preferable to discuss further the potential for use of this platform to respond quickly to infectious disease threats, rather than repeat throughout the claim of primacy. The readership of PLOS Medicine is interested in the clinical potential of new therapeutics, which the Discussion should highlight, rather than reiterate all of the results.

Please insert figures according to PLOS Medicine formatting requirements.

Please format references according to PLOS Medicine formatting requirements.

General Formatting

* Please review your text for claims of novelty or primacy (e.g. 'for the first time') and remove this language. In addition, please check that any use of statistical terms (such as trend or significant) are supported by the data, and if not please remove them.

"* Statistical reporting: Please revise throughout the manuscript, including tables and figures.

- Please report statistical information as follows to improve clarity for the reader ""22% (95% CI [13,28]; p</=)"".

- Please separate upper and lower bounds with commas instead of hyphens as the latter can be confused with reporting of negative values.

- Please repeat statistical definitions (HR, CI etc.) for each set of parentheses."

* Please replace "subject" with participant, patient, individual, or person.

* It appears that one or more study authors is affiliated with one or more of the agencies that funded the study. Thus, the statement “The funders had no role in study design, data collection and analysis, decision to publish, or preparation of the manuscript” does not apply. Please revise the Financial Disclosure accordingly, as in "[Author name] is [author's role] at [funding agency]. The funders had no other role in study design…..”

* Thank you for agreeing to make your data available. At this time, please provide the link to the data repository and accession numbers required for access.

* Please specify whether informed consent was written or oral. Please ensure that the research complies with the PLOS policy in full: https://journals.plos.org/plosmedicine/s/human-subjects-research#loc-patient-privacy-and-informed-consent-for-publication

* Please provide titles and legends for all figures and tables (including those in Supporting Information files). Please define all acronyms used in each figure or table in its corresponding legend.

* In the Kaplan-Meier curve(s) please provide the number at risk for each time interval.

* Please consider avoiding the use of red and green in order to make your figure more accessible

* Please confirm that your title complies with PLOS Medicine's style. Your title must be nondeclarative and not a question. It should begin with main concept if possible. "Effect of" should be used only if causality can be inferred, i.e., for an RCT. Please place the study design ("A randomized controlled trial," "A retrospective study," "A modelling study," etc.) in the subtitle (ie, after a colon).

* Please confirm that your abstract complies with our requirements, including format (three sections: Background, Methods and Findings, and Conclusions) and providing all the information relevant to this study type https://journals.plos.org/plosmedicine/s/submission-guidelines#loc-abstract

* Please ensure that the Introduction ends with a clear description of the study question or hypothesis.

* Please ensure that all abbreviations are defined at first use throughout the text.

* Please confirm that all numbers presented in the abstract are present and identical to numbers presented in the main manuscript text.

Clinical Trial Reporting Requirements

* Please complete the CONSORT 2025 checklist and ensure that all components of CONSORT 2025 are present in the manuscript, including how randomization was performed, allocation concealment, blinding of intervention, definition of lost to follow-up, power statement. When completing the checklist, please use section and paragraph numbers, rather than page numbers. The checklist should be included as supporting information, and should be cited in the article.

"* As your trial had to undergo important modifications in response to extenuating circumstances, please complete the CONSERVE-CONSORT checklist and provide in your Supporting Information.

When completing the checklist, please use section and paragraph numbers, rather than page numbers."

* PLOS Medicine requires that all trials be prospectively registered in one of registries recognized by WHO. Please provide information on study registration in the Methods section.

* Your trial was registered after the participants were randomized. Please explain in the paper why your trial was registered late. In your rebuttal letter, please indicate if you are conducting or have conducted any related or similar trials, and confirm that those have been registered.

* Some of the outcome measures or methods appear to differ between the submitted manuscript and the trial registry and/or protocol. Please clarify and explain the discrepancy. If the outcomes were not prespecified in the protocol, please indicate that they were post hoc and explain why they were added. Post hoc comparisons should be presented as hypothesis generating rather than conclusive.

* In accordance with ICMJE requirements, PLOS Medicine requires prospective, public registration of a data sharing plan (as part of mandatory clinical trials registration) for all clinical trials that began enrollment on or after January 1, 2019.

* The trial registration or protocol lists secondary outcomes can you please present those results as part of this manuscript, or indicate why that is not possible? If this is not possible, can you please indicate when you plan to publish those results?

* The sample size listed in the submitted manuscript and the trial registry differ. Please explain the discrepancy.

* The main analysis should be intention to treat (ie, all individuals randomized are included in the analysis in the groups to which they were originally assigned. If the study included dropouts, specify whether their data are imputed and if so using what method. Please refer to as modified ITT).

* The CONSORT flowchart should be figure 1, please revise.

* In the flow diagram, please indicate the number of individuals in each group analyzed in the ITT analysis.

* Please present the safety data for the study including numbers of specific events and whether or not adverse events are thought to be related to treatment.

* In this test of superiority it is not possible to determine that the intervention and control are equivalent, just that the intervention is not superior to the control, please update the manuscript accordingly.

* Causal language - In trials, there is usually a distinction in the language in terms of causal vs associational for primary and secondary trial outcomes. It would be beneficial to use associational language in the discussion and other sections for secondary outcomes.

* Please report your abstract according to CONSORT for abstracts, following the PLOS Medicine abstract structure (Background, Methods and Findings, Conclusions) https://www.equator-network.org/reporting-guidelines/consort-abstracts/

* Please include the clinical trial registry number in the abstract.

Per CONSORT, please note that only the primary outcome of the trial should be reported in your Abstract. Secondary outcomes should only be included in the Abstract if all secondary outcomes are fully reported. For trials that have many secondary outcomes, the Abstract should be limited to reporting the primary outcome.

Comments from Reviewers:

Reviewer #1: The authors have thoroughly addressed my comments and the comments of the other reviewers.

Reviewer #3:

I thank the authors for responding to my comments. However, I am not sure if I agree with the authors' conclusion that the change in circulating variants cannot explain the shorter duration of symptoms in the 1000mg group. As the authors mentioned, all patients in the 250mg group had Delta, whereas 6/9 patients in 500mg group had Delta and in the 1000mg group, no one had Delta infection (4 had Omicron BA.1, 4 with recombinant and 1 with unknown variant). The 1000mg group also had a shorter time to symptom disappearance and shorter median time to hospital discharge. Since XVR011 has very poor neutralising activity against Omicron BA.1 (IC50 of 9816ng/ml) compared to Delta (IC50 of 13 ng/ml), it should have a much poorer protective effect against Omicron BA.1. How can the authors attribute the reduced time of symptoms and hospitalisations in the 1000mg group to XVR011 then, if XVR011 has such poor neutralising activity against Omicron BA.1? Wouldn't this be more likely due to a change in the infecting variant (Omicron BA.1 vs Delta)?

Reviewer #4: Thank you to the authors for addressing my previous comments.

The one remaining comment I had was on my previous comment 5 which was related to the line "Statistical analysis of dose-proportionality indicated that there was no significant deviation from dose proportionality for Cmax and AUC (p-values were all >0.05) (Tables 3, S5)." which is now on lines 403-404 of the clean revised version. I see now that I misread that as Figure S5 rather than Table S5 originally. I think the statement could still be more clearly linked to what's shown in table S5 - perhaps saying in the caption of S5 that the parameters taking value 1 would mean dose proportionality?

Other than that, I don't have any additional issues to raise.

[LINK]

---

## [Editor Report · Decision Letter 3]

16 Apr 2026

Dear Dr. Herschke,

Thank you very much for re-submitting your manuscript "Characterization of VHH-Fc construct XVR011 (rimteravimab) in healthy adults and patients hospitalized for mild-to-moderate COVID-19: two Phase 1 randomized clinical trials." (PMEDICINE-D-25-00069R3) for review by PLOS Medicine.

The remaining issues that need to be addressed are in the document attached to this email. Please note that in addition to the requested changes to the manuscript, we require that you revise the data availability statement to transparently state all restrictions to data access and the process for requesting access.

Please submit a clean version of the paper as the main article file. A version with changes marked must also be uploaded as a marked up manuscript file.

We expect to receive your revised manuscript within 4 days. Please email us (plosmedicine@plos.org) if you have any questions or concerns.

We look forward to receiving the revised manuscript by Apr 23 2026 11:59PM.

Sincerely,

Alison Farrell, Ph.D.

Senior Editor

PLOS Medicine

plosmedicine.org

---

## [Editor Report · Decision Letter 4]

23 Apr 2026

Dear Dr Herschke,

On behalf of my colleagues and the Academic Editor, Matthias Egger, I am pleased to inform you that we have agreed to publish your manuscript "Characterization of the VHH-Fc construct rimteravimab in healthy adults and patients hospitalized for mild-to-moderate COVID-19: two Phase 1 randomized clinical trials." (PMEDICINE-D-25-00069R4) in PLOS Medicine.

We also ask you to revise your files to address the following issues:

Please add the funder URLs to the manuscript metadata.

In the Methods section that identifies the IRBs, please add a callout to the IRB approval numbers in the SI.

In the Methods, explain what the placebo is in the phase 1a trial.

Line 70: please correct " IgG1, silenced for Fc effector functions-. " to " IgG1 Fc, silenced for Fc effector functions." [please note addition of 'Fc' and deletion of hyphen after 'functions']'.

Line 77: after '1000 mg' add 'of XVR011

Line 81: 'was evaluated as a secondary objective' please correct to 'were evaluated as secondary objectives'

Line 210: participants is written twice. Please delete one instance.

Line 216: Insert period after '(IV)'.

Line 404: should 'investigator' be plural? The investigator is not identified.

Line 766 in Table 4 legend: 'response' should be 'responses'

Line 769: 'are' should be 'is'

In the Methods, authors need to include callouts to the separate protocols for each phase 1 study.

Please state in the Methods whether there was or was not an independent Data Safety Monitoring Board.

When completing the CONSORT checklist, please use section and paragraph numbers, rather than page numbers. Please provide a revised CONSORT checklist.

In the SI, please add an additional sentence clarifying what is meant by 'simple randomization'. Please refer to the CONSORT guidelines for clarification.

Authors Viki Bockstal, Angélique Boerboom and Natalia Gaibu have not completed the questionnaire (letter were re-sent to them).

Authors Manuela Rinaldi and Salah Hadi have incorrectly declared no competing interests. Both authors have competing interest as employees (or former employees) of companies. Please ask the authors to correct their statements in the questionnaire.

PRESS

Sincerely,

Alison Farrell, Ph.D.

Senior Editor

PLOS Medicine